# LEARNING ABSTRACT MODELS FOR LONG-HORIZON EXPLORATION

## ABSTRACT

In high-dimensional reinforcement learning settings with sparse rewards, performing effective exploration to even obtain any reward signal is an open challenge. While model-based approaches hold promise of better exploration via planning, it is extremely difficult to learn a reliable enough Markov Decision Process (MDP) in high dimensions (e.g., over $10^{100}$ states). In this paper, we propose learning an *abstract MDP* over a much smaller number of states (e.g., $10^5$), which we can plan over for effective exploration. We assume we have an abstraction function that maps concrete states (e.g., raw pixels) to abstract states (e.g., agent position, ignoring other objects). In our approach, a *manager* maintains an abstract MDP over a subset of the abstract states, which grows monotonically through targeted exploration (possible due to the abstract MDP). Concurrently, we learn a *worker* policy to travel between abstract states; the worker deals with the messiness of concrete states and presents a clean abstraction to the manager. On three of the hardest games from the Arcade Learning Environment (MONTEZUMA'S REVENGE, PITFALL!, and PRIVATE EYE), our approach outperforms the previous state-of-the-art by over a factor of 2 in each game. In PITFALL!, our approach is the first to achieve superhuman performance without demonstrations.[1]

## 1 INTRODUCTION

Exploration is a key bottleneck in high-dimensional, sparse-reward reinforcement learning tasks. Random exploration (e.g., via epsilon-greedy) suffices when rewards are abundant (Mnih et al., 2015), but when rewards are sparse, it can be difficult for an agent starting out to even find *any* positive reward needed to bootstrap learning. For example, the infamously difficult game MONTEZUMA'S REVENGE from the Arcade Learning Environment (ALE) (Bellemare et al., 2013) contains over $10^{100}$ states and requires the agent to go thousands of timesteps without receiving reward. Performing effective exploration in this setting is thus an open problem; without demonstrations, even state-of-the-art intrinsically-motivated RL agents (Bellemare et al., 2016; Ostrovski et al., 2017; Tang et al., 2017) achieve only about one-tenth the score of an expert human (Hester et al., 2018).

In this paper, we investigate model-based reinforcement learning (Kearns & Singh, 2002) as a potential solution to the exploration problem. The hope is that with a model of the state transitions and rewards, one can perform *planning* under the model to obtain a more informed exploration strategy. However, as the model being learned is imperfect, errors in the model *compound* (Talvitie, 2014; 2015) when planning over many time steps. Furthermore, even if a perfect model were known, in high-dimensional state spaces (e.g. over $10^{100}$ states), planning—computing the optimal policy (e.g. via value iteration)—is intractable. As a result, model-based RL has had limited success in high-dimensional settings (Talvitie, 2014). To address this, some prior work has focused on learning more accurate models by using more expressive function approximators (Nagabandi et al., 2018), and learning local models (Levine & Koltun, 2013; Zhang et al., 2018). Others have attempted to robustly use imperfect models by conditioning on, instead of directly following, model-based rollouts (Weber et al., 2017), frequently replanning, and combining model-based with model-free approaches (Abbeel et al., 2006; Sutton, 1990). However, none of these techniques offer a fundamental solution.

Instead of directly learning a model over the concrete state space, we propose an approach inspired by hierarchical reinforcement learning (HRL) (Sutton et al., 1999; Vezhnevets et al., 2017; Oh et al.,

---

[1] Videos of our trained agent: `https://sites.google.com/view/abstract-models/home`

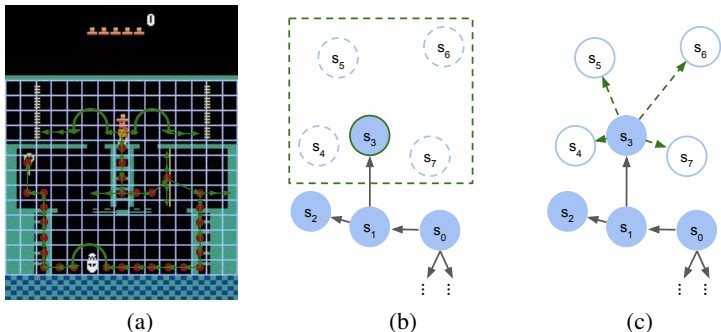

Figure 1: (a) Illustration of the abstract MDP on MONTEZUMA'S REVENGE. We have superimposed a white grid on top of the original game. At any given time, the agent is in one of the grid cells – each grid cell is an abstract state. In this example, the agent starts at the top of a ladder (yellow dot). The worker then navigates transitions between abstract states (green arrows) to follow a plan made by the manager (red dots). (b) Circles represent abstract states. Shaded circles represent states within the known set. The manager navigates the agent to the fringe of the known set ($s_3$), then randomly explores with $\pi^d$ to discover new transitions near $s_3$ (dotted box). (c) The worker extends the abstract MDP by learning to navigate to the newly discovered abstract states (dotted arrows).

2017), and learn a model over a much smaller abstract state space. Specifically, we assume we have a state abstraction function (Li et al., 2006; Singh et al., 1995; Dietterich, 1998), which maps a high-dimensional concrete state (e.g. all pixels on the screen) to a low-dimensional abstract state (e.g. the position of the agent). We then aim to learn an (abstract) Markov Decision Process (MDP) over this abstract state space as follows: A *manager* maintains an (abstract) MDP over a subset of all possible abstract states which we call the known set, which is grown over time. The crucial property we enforce is that this abstract MDP is highly accurate and near deterministic on the known set, so we can perform planning without suffering from compounding errors, and do it efficiently since we are working with a much smaller number of abstract states. Concurrently, we learn a *worker policy* that the manager uses to transition between abstract states. The worker policy has access to the concrete states; its goal is to hide the messy details of the real world from the manager (e.g., jumping over monsters) so that the manager has a much simpler planning problem (e.g., traversing between two locations). In our implementation, the worker keeps an inventory of *skills* (i.e., options (Sutton et al., 1999)), each of which is driven by a deep neural network; the worker assigns an appropriate skill for each transition between abstract states. In this way, the worker does not "forget" (Kirkpatrick et al., 2017), and we ensure monotonic progress in learning the abstract MDP. This abstract MDP, which enables us to efficiently explore via planning, is a key difference between our work and previous HRL work (e.g., (Bacon et al., 2017; Vezhnevets et al., 2017)), which also learn skills and operate on latent abstract state spaces but without forming an MDP.

We evaluate our approach on three of the most challenging games from the ALE (Bellemare et al., 2013): MONTEZUMA'S REVENGE, PITFALL!, and PRIVATE EYE. In all three domains, our approach achieves more than 2x the reward of prior non-demonstration state-of-the-art approaches. In PITFALL!, we are the first to achieve superhuman performance without demonstrations, surpassing the prior state-of-the-art by over 100x. Additionally, since our approach is model-based, we can generalize to new rewards without re-training, as long as the reward function is a function of the abstract states. When evaluated on a new reward function never seen during training, our approach achieves over 3x the reward of prior state-of-the-art methods explicitly trained on the new rewards.

## 2 APPROACH OVERVIEW

We assume the world is an unknown episodic finite-horizon MDP with (concrete) states $x \in \mathcal{X}$ and actions $a \in \mathcal{A}$. We further assume we have a simple predefined *state abstraction function* mapping concrete states $x$ to abstract states $s = \phi(x)$. In MONTEZUMA'S REVENGE, for instance, a concrete state contains the pixels on the screen, while the corresponding abstract state contains the agent's

position and inventory (Figure 1). We assume that reward only depends on the abstract states: taking any action transitioning from concrete state $x$ to $x'$ leads to reward $R(\phi(x), \phi(x'))$.

Model-based approaches promise better exploration via planning, but struggle in high-dimensional state spaces due to compounding errors and computational intractability. To avoid these problems, we propose to construct and operate on a low-dimensional representation of the world consisting of abstract states, which we call the *abstract MDP* (we refer to the original MDP, the world, as the concrete MDP). Then we plan in the abstract MDP. Concretely, the abstract MDP consists of:

- The *state space* is a subset of all abstract states, which we call the *known set $\mathcal{S}$*, consisting of abstract states that can be reliably reached from the initial abstract state via actions in the action set. Over time, we monotonically grow the *known set*, which initially only contains the starting abstract state.

- The *action set* comprises of calling the *worker* policy $\pi^w(a|x, (s, s'))$ on transitions $(s, s')$ from the current abstract state $s$ to a nearby abstract state $s'$. When called on a transition $(s, s')$, the worker navigates from $s$ to $s'$ by taking concrete actions $a$ conditioned on the current concrete state $x$. The worker abstracts away the messiness of the underlying concrete MDP so that all other parts of the system can operate on the abstract MDP by calling the worker. We denote calling the worker on transition $(s, s')$ as the action $\mathrm{go}(s, s')$.

- The *transition dynamics* of calling action $\mathrm{go}(s, s')$ at abstract state $s$ are defined by the original MDP: i.e., if the worker takes a concrete trajectory $x_0, a_0, x_1, a_1, \cdots, x_T$, the resulting abstract state is $\phi(x_T)$. The *rewards* for transitioning from $s$ to $s'$ are the rewards in the concrete MDP $R(s, s')$, which only depend on the abstract states by assumption.

The core idea behind our approach is to construct the abstract MDP (Section 3) by growing the action set (training the worker), which in turn grows the known set. At each point in time, the *manager* maintains the known set and (accurate, to avoid compounding errors) estimates of the reward and transition dynamics of the abstract MDP. With these dynamics estimates, the manager can solve the abstract MDP at all times (e.g., by value iteration), since the abstract MDP is small. As the abstract MDP grows, it captures more and more of the concrete MDP, enabling the manager to recover a better and better policy via planning. Ultimately, the known set of the abstract MDP contains all abstract states, enabling the manager to recover a high-reward policy.

As the manager constructs the abstract MDP, the abstract MDP maintains two key properties:

- The action set of the abstract MDP consists of only *reliable* actions, actions $\mathrm{go}(s, s')$ that transition from abstract state $s$ to abstract state $s'$ with probability at least $1 - \delta$ for some small $\delta$ to avoid compounding uncertainty. This enables the manager to reach any abstract state in the known set with high probability. To simplify notation, the manager estimates the success rate $P(s, s')$ of action $\mathrm{go}(s, s')$ instead of the full dynamics $P(\bullet|\mathrm{go}(s, s'), s)$, treating the (vanishingly small) fraction of failures equally.

- The action set and known set of the abstract MDP grow *monotonically*. Since the action set only contains reliable transitions, a key danger is if learning new reliable transitions (adding new actions) causes the worker to forget already learned transitions (removing actions), stalling progress. We opt for a non-parametric approach, where the worker learns a *skill* (neural subpolicy) for each transition, reusing skills when possible. When a worker learns to reliably traverse a transition, it freezes the corresponding skill's parameters.

## 3 Constructing the Abstract MDP

The manager's goal is to fully construct the abstract MDP so that the known set contains all abstract states. Then, it can compute a high-reward policy on the concrete MDP via planning on the abstract MDP. To construct the abstract MDP, the manager adds new actions to the abstract MDP: training the worker to reliably traverse new transitions (driving the transition success rates toward 1). Concretely, the manager discovers new transitions, trains the worker on these transitions, and updates its dynamics estimates using Algorithm 1. On each episode, the manager either chooses to discover new transitions via randomized exploration (Section 3.1) or trains the worker. This is done by constructing a prioritized list of *exploration goals*, where each goal is either a transition to learn

---

**Algorithm 1** MANAGER

1: **while** abstract MDP not fully constructed **do**
2:     Compute a set of candidate exploration goals $C$
3:     ($c \in C$ is either a transition $(s, s')$ or an abstract state $s$)
4:     Score all candidates and select highest priority candidate $c$
5:     Compute a plan $(s_0, s_1), (s_1, s_2), \cdots, (s_{T-1}, s_T = s)$ with model
6:     **for** $t = 1$ to $T$ **do**
7:         Call worker to navigate transition $(s_{t-1}, s_t)$
8:     **if** $c$ is a transition $(s, s')$ to learn **then**
9:         LEARNWORKER$(s, s')$
10:    **else** $c$ is an abstract state $s$ to explore
11:        DISCOVERTRANSITIONS()

---

or an abstract state to find nearby transitions (Section 3.2). Upon selecting the highest-priority goal (e.g., the transition $(s, s')$), the manager navigates to the relevant abstract state (e.g., $s$) by planning with its dynamics models (e.g., the plan $\mathrm{go}(s_0, s_1), \mathrm{go}(s_1, s_2), \cdots, \mathrm{go}(s_{T-1}, s_T = s)$), executing the plan, and then calling the worker on or randomly exploring from the selected goal. Finally, the manager updates its dynamics models. It uses a sliding window estimate of the past $N_{transition}$ worker attempts of traversing transition $(s, s')$ for the transition dynamics, and updates its reward estimate of a transition $(s, s')$ as the reward accumulated by the first successful traversal of $(s, s')$. When a transition $(s, s')$ becomes reliable (i.e., the dynamics $T(s, s')$ exceeds the threshold $1 - \delta$), the manager adds $\mathrm{go}(s, s')$ to the action set of the abstract MDP and adds $s'$ to the known set.

## 3.1 DISCOVERING NEW TRANSITIONS

For the manager to train the worker on new transitions, it must first discover new transitions. To discover new transitions, the manager navigates to an *exploration candidate*: an abstract state $s$. Then, it performs randomized exploration to discover new transitions $(s, s')$ to nearby abstract states $s'$. As exploration candidates, the manager simply chooses the abstract states in the known set that have been explored fewer than $N_{visit}$ times: i.e., $n(s) < N_{visit}$, where $n(s)$ is the number of times the manager has explored abstract state $s$ for nearby transitions. Effectively, the manager assumes that when $n(s) \geq N_{visit}$, all nearby transitions $(s, s')$ have been already discovered.

Concretely, at an abstract state $s$, the manager finds nearby transitions by following a simple policy $\pi^d(a_t | x_{0:t}, a_{0:t-1})$ for $T_d$ timesteps (Algorithm 3). The policy $\pi^d$ outputs randomized concrete actions $a_t$ conditioned on the past concrete states $x_{0:t}$ and past concrete actions $a_{0:t-1}$, where $\phi(x_0)$ is the abstract state $s$ it was initially invoked. During those $T_d$ timesteps, the manager records the transitions and rewards it observes: $(\phi(x_0), r_0, \phi(x_1)), \cdots (\phi(x_{T-1}), r_{T-1}, \phi(x_T))$, using the rewards to update its rewards model and the transitions as candidates for the worker to learn. Additionally, if it ends in another exploration candidate (i.e., $n(s) < N_{visit}$) after exploring for $T_d$ timesteps, it simply continues exploring for another $T_d$ timesteps.

The simplest possible policy for the $\pi^d$ is to uniformly sample a concrete action at each timestep. However, we found that this inadequately discovered new transitions, because it would often perform useless action sequences (e.g., left, right, left, right). Instead, we use a simple method for $\pi^d$ to commit to an exploration direction. At each timestep, the $\pi^d$ uniformly samples a concrete action and a number between 1 and $T_{repeat}$, and repeats the action the sampled number of times.

## 3.2 CHOOSING AN EXPLORATION GOAL

**Exploration goals.** The manager selects an exploration goal from the set of all candidate exploration goals, consisting of exploration candidates for the transition discovery and candidate transitions for the worker to learn. The exploration candidates are just the abstract states in the known set with $n(s) < N_{transition}$. The candidate transitions are the transitions discovered by the manager. In addition, the worker imposes a heuristic on its transition learning process in order to preserve the Markov property of the abstract MDP (Section 4), which sometimes makes it impossible for the worker to learn a transition. To avoid getting stuck, as candidates for the worker, the manager also considers "long-distance" transitions: $(s, s')$ pairs for which the manager did not directly transition from $s$ to $s'$, but indirectly did so through a sequence of intermediate states

---

**Algorithm 2** LEARNWORKER$(s, s', x_0)$

---

**Input:** a transition $(s, s')$ to learn, called at concrete state $x_0$ with $\phi(x_0) = s$
 1: Set worker horizon $H = d(s, s') \times H_{\text{worker}}$
 2: Choose $a_0 \sim \pi^{\text{w}}(x_0, (s, s')) = \pi_{\mathcal{I}(s,s')}(x_0, s')$
 3: **for** $t = 1$ to $H$ **do**
 4:     Observe $x_t$
 5:     Compute worker intrinsic reward $r_t = R_{(s,s')}(x_t | s')$
 6:     Update worker on $(x_{t-1}, a_{t-1}, r_t, x_t)$
 7:     Choose $a_t \sim \pi^{\text{w}}(x_t, (s, s')) = \pi_{\mathcal{I}(s,s')}(x_t, s')$
 8: Compute success = $\mathbf{1}[r_1 + \cdots + r_H \geq R_{\min}]$
 9: Update transition model $P(s, s') \leftarrow$ success rate of past $N_{transition}$ attempts
10: **if** $P(s, s') \geq 1 - \delta$ **then**
11:     Freeze worker's skill $\pi_{\mathcal{I}(s,s')}$

---

$(s_0 = s, s_1, \cdots, s_T = s')$. Letting $d(s, s')$ be the length of the shortest such path, the manager considers all pairs $(s, s')$ for which $d(s, s') \leq d_{max}$.

**Priority scores of the exploration goals.** The manager must choose exploration goals in *some* order. Our theoretical results (Section 5) hold for any priority function that eventually chooses all exploration goals. In our implementation, the manager prioritizes the easiest exploration goals (enabling the fastest growth of the action set), and the most useful goals (goal that either lead to more reward or enable the worker to learn new transitions).

Concretely, the manager heuristically computes the easiness of a learning transition $(s, s')$ as $e(s, s') = \lambda_1 n_{succ}(s, s') - n_{fail}(s, s') - d(s, s')^2$, where $n_{succ}$ is the number of times the worker has successfully traversed $(s, s')$ and $n_{fail}$ is the number of times the worker has failed in traversing $(s, s')$. Intuitively, both 1) succeeding more and failing less and 2) shorter transitions, requiring fewer timesteps indicate easier to learn transitions. Similarly, for an abstract state $s$, the manager computes the easiness of discovering new neighboring transitions as $e(s) = -n(s)$ since abstract states that have been explored less are more likely to have undiscovered transitions. The manager heuristically computes the usefulness of a learning a transition $(s, s')$ as $u(s, s') = \lambda_2 \mathbb{I}_{new} + R(s_0, s)$, where $\mathbb{I}_{new}$ is an indicator that is 1 if there is an outgoing transition $(s', s'')$ and no current candidate transitions end in $s''$. $R(s_0, s)$ is the reward achieved by navigating from the initial abstract state $s_0$ to $s$. If $\mathbb{I}_{new}$ is 1, then learning $(s, s')$ opens new candidate transitions for the worker to learn, indicating that learning $(s, s')$ is useful. For an abstract state $s$, the manager computes the usefulness just as $u(s) = \lambda_3 + R(s_0, s)$. The $\lambda_3$ constant accounts for how much more or less useful discovering new transitions is compared to learning new transitions.

To prioritize exploration goals, the manager uniformly switches between two priority functions. The first priority function simply equals the easiness plus usefulness: $e + u$, and the second priority function is the same, but without the reward term in $u$, to avoid falling into a local reward maximum.

## 4 LEARNING THE WORKER POLICY

The worker forms the action set of the abstract MDP by learning many subtasks of the form: navigate from abstract state $s$ to $s'$. It does this while maintaining the three properties: 1) the worker reliably (with high probability) traverses $(s, s')$ for each action go$(s, s')$ in the abstract MDP; 2) the action set grows monotonically, so learning new transitions never causes the worker's old transitions to become unreliable; and 3) the worker learns transitions in a way that preserves the Markov property.

While it is possible to learn a single policy for all transition subtasks, it is tricky to satisfy 2), since learning new transitions can have deleterious effects on previously learned transitions. Instead, the worker maintains an inventory of *skills* (Section A.3), where each transition is learned by a single skill, sharing the same skill amongst many transitions when possible. The worker uses these skills to form the action set of the abstract MDP following Algorithm 2: When the manager calls the worker on a transition $(s, s')$, the worker selects the appropriate skill from the skill inventory and begins an episode of the subtask of traversing $s$ to $s'$ (Section 4.2). During the skill episode, the skill receives intrinsic rewards, and is declared to have successfully completed the subtask if it meets the worker's *holding heuristic*, which heuristically maintains 3) by ensuring the worker can always control the

abstract state. If at the end of the skill episode, the success rate of the worker traversing $(s, s')$ exceeds the reliability threshold $1 - \delta$, the action $go(s, s')$ is added to the abstract MDP.

## 4.1 SKILL REPOSITORY

The worker's *skill inventory* $\mathcal{I}$ indexes skills so that the skill at index $\mathcal{I}(s, s')$ reliably traverses transition $(s, s')$. Each skill is a goal-conditioned subpolicy $\pi_{\mathcal{I}(s,s')}(a|x, s')$, which produces concrete actions $a$ conditioned on the current concrete state $x$ and the goal abstract state $s'$. When the worker traverses a transition $(s, s')$, it calls on the corresponding skill until the transition is traversed: i.e., $\pi^w(a|x, (s, s')) = \pi_{\mathcal{I}(s,s')}(a|x, s')$.

When learning a new transition $(s, s')$, the worker first tries to reuse its already learned skills from the skill inventory. For each skill $\pi_i$ in the skill inventory, it measures the success rate of $\pi_i$ on the new transition $(s, s')$ over $N_{transition}$ attempts. If the success rate exceeds the reliability threshold $1 - \delta$ for any skill $\pi_i$, it updates the skill repository to reuse the skill: $\mathcal{I}(s, s') \leftarrow \pi_i$. Otherwise, if no already learned skill can reliably traverse the new transition, the worker creates a new skill and trains it to navigate the transition by optimizing intrinsic rewards during skill episodes (Section 4.2).

## 4.2 WORKER SUBTASK

Given a transition $(s, s')$, the worker's subtask is to navigate from abstract state $s$ to abstract state $s'$. Each episode of this subtask consists of $d(s, s') \times H_{worker}$ timesteps (longer transitions need more timesteps to traverse), where the reward at each timestep is $R_{(s,s')}(x_t) = 1$ if the skill has successfully reached the end of the transition ($\phi(x_t) = s'$) and 0 otherwise. These episodes additionally terminate if the main episode terminates or if the manager receives negative environment reward.

When solving these subtasks to construct the action set of the abstract MDP, the worker must be careful not to violate the Markov property. In particular, the concrete state may contain some history-dependent information lost due to the state abstraction function. For example, consider the task of jumping over a dangerous hole, consisting of three abstract states: $s_1$ (the cliff before the hole), $s_2$ (the air above the hole), and $s_3$ (the solid ground on the other side of the hole). The worker might incorrectly assume that it can reliably traverse from $s_1$ to $s_2$ by simply walking off the cliff. But adding this as a reliable transition to the abstract MDP causes a problem: there is now no way to successfully traverse from $s_2$ to $s_3$ due to missing history-dependent information in the abstract state (i.e., the way the worker navigated $s_1$ to $s_2$), violating the Markov property.

On navigating a transition $(s, s')$, the worker avoids this problem by navigating to $s'$ and then checking for history-dependent consequences with the *holding heuristic*. The worker assumes that if its navigation of $(s, s')$ changed unobserved parts of the state, then those changes would eventually cause the abstract state to change (e.g., in the example, the worker would eventually hit the bottom and die). Consequently, if the worker can stay in $s'$ for many timesteps, then it did not significantly unobserved parts of the state. This corresponds to only declaring the episode as a success if the worker accumulates at least $R_{hold}$ reward (equivalent to being in $s'$ for $R_{hold}$ timesteps).

Any RL algorithm can be used to represent and train the skills to perform this subtask. We choose to represent each skill as a Dueling DDQN (van Hasselt et al., 2016; Wang et al., 2016). For faster training, the skills use self-imitation (Oh et al., 2018) to more quickly learn from previous successful episodes, and count-based exploration similar to (Bellemare et al., 2016) to more quickly initially discover skill reward. Since the skill inventory can contain many skills, we save parameters by occasionally using pixel-blind skills. Appendix A.3 fully describes our skill training and architecture.

## 5 FORMAL ANALYSIS

We are interested in the *sample complexity* (Kakade et al., 2003), the number of samples required to learn a policy that achieves reward close to the optimal policy, with high probability. Standard results in the tabular setting (e.g., MBIE-EB (Strehl & Littman, 2008)) guarantee learning a near-optimal policy, but require a number of timesteps polynomial in the size of the state space, which is effectively vacuous in the deep RL setting, where state spaces can be exponentially large (e.g.,

$> 10^{100}$ states). In contrast, our approach is able to use time and space polynomial in the size of the abstract state space, which is exponentially smaller, by operating on the abstract MDP.

Formally, assuming that our neural network policy class is rich enough to represent all necessary skills, with high probability, our approach can learn a near-optimal policy on a subclass of MDPs in time and space polynomial in the size of the abstract MDP (details in Appendix C). The key intuition is that instead of learning a single task where the time horizon is the length of the game, our approach learns many subtasks where the time horizon is the number of steps required to navigate from one abstract state to another. This is critical, as many deep RL algorithms (e.g. $\epsilon$-greedy) require a number of samples exponential in the time horizon to solve a task.

## 6 EXPERIMENTS

Following (Aytar et al., 2018), we empirically evaluate our approach on three of the most challenging games from the ALE (Bellemare et al., 2013): MONTEZUMA'S REVENGE, PITFALL!, and PRIVATE EYE. We do not evaluate on simpler games (e.g., Breakout), because they are already solved by prior state-of-the-art methods (Hessel et al., 2017) and do not require sophisticated exploration. We use the standard ALE setup (Appendix A) and end the episode when the agent loses a life. We report rewards from periodic evaluations every 4000 episodes, where the manager plans for optimal reward in the currently constructed abstract MDP. We average our approach over 4 seeds and report 1 standard deviation error bars in the training curves. Our experiments use the same set of hyperparameters (Appendix A.1) across all three games, where the hyperparameters were exclusively and minimally tuned on MONTEZUMA'S REVENGE.

In all three games, the state abstraction function uses the RAM state, available through the ALE simulator, to extract the bucketed location of the agent and the agent's current inventory. Roughly, this distinguishes states where the agent is in different locations or has picked up different items, but doesn't distinguish states where other details differ (e.g. monster positions or obstacle configurations). Notably, the abstract state function does not specify what each part of the abstract state means, and the agent does not know the entire abstract state space beforehand. We describe the exact abstract states in Appendix A.2.

### 6.1 MAIN RESULTS

Among the many deep RL approaches, in each game, we compare with the prior non-demonstration state-of-the-art approach, which use prior knowledge comparable to our RAM information:

- In MONTEZUMA'S REVENGE, we compare with *SmartHash* (Tang et al., 2017), a count-based exploration approach which estimates state visit counts with a hash-based density model and provides intrinsic reward to revisit states with low visit counts. Like our approach, *SmartHash* also uses RAM state information. It hashes each state to a hand-selected subset of the RAM state and maintains visit counts on the hashes.[2]

- In PITFALL!, we compare with *SOORL* (Keramati et al., 2018), a planning approach which requires prior knowledge to extract the objects on the screen. *SOORL* is the only prior non-demonstration approach to achieve positive reward in PITFALL!, but requires extensive engineering (much stronger than RAM state info) to identify and extract all objects. Once *SOORL* has access to the objects, it learns in very few frames since data from similar objects can be pooled in learning. Consequently, in our training curves, we report its final average performance over 100 runs, as well as its final best performance over 100 runs,

- In PRIVATE EYE, we compare with another count-based exploration method, *DQN-PixelCNN* (Ostrovski et al., 2017), which uses a pixel-based density model to estimate state visitation counts. We compare with the results reported in Ostrovski et al. (2017). *DQN-PixelCNN* uses less prior knowledge than our approach, but we compare with it because it achieves the previous non-demonstration state-of-the-art results.

- In all three games, we compare with *AbstractStateHash*, which performs count-based exploration identical to *SmartHash*, but uses the same RAM information as our approach.

[2]The results from *SmartHash* and *AbstractStateHash* are obtained by running code generously provided by the original authors (Tang et al., 2017).

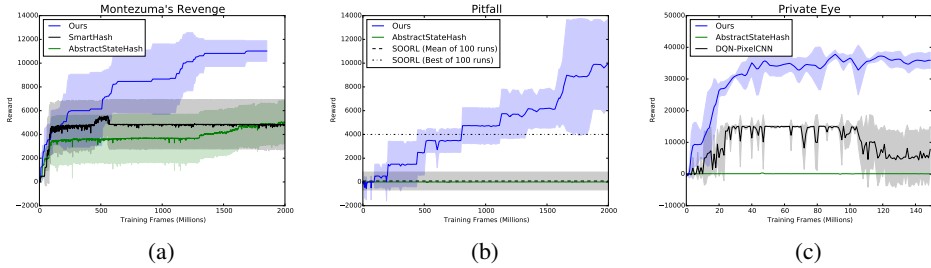

(a)          (b)          (c)

Figure 2: Comparison of our approach with the prior non-demonstration state-of-the-art approaches in MONTEZUMA'S REVENGE, PITFALL!, and PRIVATE EYE. Our approach achieves more than double the reward of prior state-of-the-art approaches in all three.

| Task | Ours (Best Seed) | Ours (Worst Seed) | Prior SOTA (Best Seed) |
|---|---|---|---|
| MONTEZUMA'S REVENGE | **12500** | *10500* | 6600 (*SmartHash*) |
| PITFALL! | **15000** | *6000* | 4000 (*SOORL*) |
| PRIVATE EYE | **40100** | 36200 | *39000 (DQN-PixelCNN)* |

Table 1: Comparison between our worst performing seed with the best performing seed of the prior-state-of-the-art. Our worst performing seed outperforms the prior best on MONTEZUMA'S REVENGE and PITFALL! and performs comparably to the prior best on PRIVATE EYE. Our best performing seed achieves new peak rewards.

Figure 2 shows the main results. *AbstractStateHash* matches the prior state-of-the-art on MONTEZUMA'S REVENGE, but performs relatively poorly on PRIVATE EYE and PITFALL!. This suggests both that prior state-of-the-art methods do not effectively leverage the state abstraction function, and that the state abstraction function does not trivialize the learning problem.

In MONTEZUMA'S REVENGE, after 2B training frames, our approach achieves a final average reward of 11020, more than doubling the average reward of *SmartHash*: 5001. Our approach achieves higher average reward than *SmartHash* at every point along the training curves and continues to learn even late into training, while *SmartHash* plateaus (Appendix B.3 presents more results on the ability of our approach to continue to learn without plateauing).

Our approach is the first non-demonstration approach to achieve superhuman performance on PIT-FALL!, achieving a final average reward of 9959.6 after 2B frames of training, compared to average human performance: 6464 (Pohlen et al., 2018). In addition, our approach achieves more than double the reward of *SOORL*, which achieves a maximum reward of 4000 over 100 seeds and a mean reward of 80.52, and even significantly outperforms *Ape-X DQfD* (Pohlen et al., 2018), which uses high-scoring expert demonstrations during training to achieve a final mean reward of 3997.5.

In PRIVATE EYE, our approach achieves a mean reward of 35636.1, more than double the reward of *DQN-PixelCNN*, which achieves 15806.5. Our approach performs even better, approaching human performance, if we change a single hyperparameter (Appendix B.4).

**Stability.** Recent work (Henderson et al., 2017) has drawn attention to the instability of deep RL results. To highlight the stability of our results, we compare our worst performing seed against the prior state-of-the-art's best performing seed in Table 1. Even our worst seed outperforms the mean performance of the prior state-of-the-art approaches. In addition, our worst seed is competitive with the highest previously reported rewards in each of the games, significantly outperforming the previous high in MONTEZUMA'S REVENGE and PITFALL!, and narrowly performing worse than the previous high in PRIVATE EYE. Even in PRIVATE EYE, while *DQN-PixelCNN* achieves 39000 reward on its best single episode across all seeds, none of its seed consistently achieves more than 15806.5 reward over many episodes. In contrast, our worst seed consistently obtains 36200 reward. Furthermore, our best seeds achieve new peak performances in each of the games.

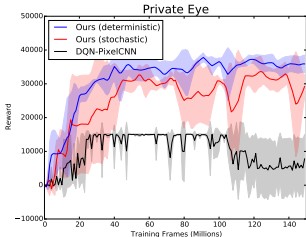

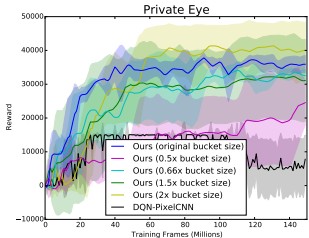

Figure 3: Our method continues to out-perform the prior state-of-the-art on the stochastic version of PRIVATE EYE.

Figure 4: Our method outperforms the prior state-of-the-art on a wide range of granularities of the state abstraction function.

## 6.2 GENERALIZATION TO NEW TASKS

By using the abstract MDP, our approach can quickly generalize to new tasks in the same environ-ment that were not seen during training. It does this by revisiting the transitions in the abstract MDP to update the rewards model with the newly observed reward. We study the ability of our approach to do this. After our approach completes training on the original reward function on MONTEZUMA'S REVENGE, we evaluate its performance on three new reward functions, allowing our approach to interact with the environment for an additional 1M frames to observe each new reward function. We compare with *SmartHash*, trained from scratch directly on the new reward functions for 1B frames. Even when evaluated on an unseen reward function, our approach achieves about 3x as much reward as *SmartHash*, which is directly trained on the new reward function. This suggests that our approach can generalize to new tasks with the same dynamics. We describe the details in Appendix B.2.

## 6.3 ROBUSTNESS TO STOCHASTICITY

We additionally evaluate the performance of our approach on the recommended (Machado et al., 2017) form of ALE stochasticity (sticky actions 25% of the time) on PRIVATE EYE (selected because it requires the fewest frames for training). Figure 3 compares the performance of our method on the stochastic version of PRIVATE EYE with the performance of our method on the deterministic version of PRIVATE EYE. Performance degrades slightly on the stochastic version, because the worker's skills become harder to learn. However, both versions outperform the prior state-of-the-art *DQN-PixelCNN*, and the worker is able to successfully abstract away stochasticity from the manager in the stochastic version of the game so that the abstract MDP remains near-deterministic.

## 6.4 VARYING THE GRANULARITY OF THE ABSTRACT STATE

To see how easy it is to create a high-performing state abstraction function on new tasks, we study the robustness of our approach to state abstraction functions of varying degrees of coarseness on PRIVATE EYE. Our state abstraction function buckets the agent's $(x, y)$ coordinates. We vary the coarseness of the abstraction function by varying the bucketing size: increasing the bucketing size results in fewer, coarser abstract states.

We report results in Figure 4 on five different bucket sizes obtained by scaling the original bucket size by $\frac{1}{2}, \frac{2}{3}, 1, \frac{3}{2}$, and 2. To adjust for the updated bucket sizes, we also scale the worker's skill episode horizon $H_{worker}$ by the same value. Our method outperforms the prior state-of-the-art approach *DQN-PixelCNN* across the entire range of bucket sizes, suggesting that our approach does not require a highly tuned state abstraction function.

## 7 RELATED WORK

Exploration in tabular settings is well-understood via optimism in the face of uncertainty (OFU) (Brafman & Tennenholtz, 2002; Strehl et al., 2009) and posterior sampling (Osband et al., 2013; Osband & Roy, 2016). OFU methods (Brafman & Tennenholtz, 2002; Strehl & Littman, 2005;

Jaksch et al., 2010) achieve provably near-optimal policies in polynomial time by providing reward bonuses for exploring regions of uncertainty. Nevertheless, despite recent stronger optimality bounds (Azar et al., 2017; Dann et al., 2017), these methods do not scale to the deep RL setting, where the state space is prohibitively large. Bellemare et al. (2016); Tang et al. (2017); Ostrovski et al. (2017) similarly apply optimism reward bonuses in the case of high-dimensional state spaces, empirically improving exploration. However, these methods no longer guarantee optimality, and can suffer from insufficient exploration because they reactively seek infrequently visited states, whereas our manager proactively seeks new abstract states.

While model-based RL succeeds in the tabular setting (Strehl et al., 2009) and in tasks with relatively small (e.g., $< 100$ dimensions) state spaces (Nagabandi et al., 2018), it has little success on tasks with exponentially large state spaces (e.g., pixel inputs). Prior work (Oh et al., 2015) that learns models in these large state spaces suffers from compounding errors (Talvitie, 2014), where long-term predictions become extremely inaccurate. To make matters worse, while prior model-based works (Weber et al., 2017; Ha & Schmidhuber, 2018) succeed on relatively dense reward tasks, model-based planning methods (e.g., value iteration) can be computationally intractable in long-horizon, sparse-reward tasks, even when a perfect model is known. To circumvent these problems, our work learns an abstract MDP consisting of an exponentially smaller abstract state space and learned skills. Oh et al. (2017) similarly learns a model over abstract states and skills, but uses manually engineered skills, whereas ours are learned.

Our work relates to prior work on hierarchical reinforcement learning (HRL), which also operates on abstract states (Dietterich, 2000; Li et al., 2006) with learned skills or subgoals (Schmidhuber, 1993; Singh et al., 1995; Vezhnevets et al., 2017; Bacon et al., 2017). However, a key difference is that our work constructs the abstract MDP, enabling us to perform targeted exploration via planning and rely on the Markov property to avoid exponentially large state histories. In contrast, the abstract states and skills in these other works do not meet such useful structural properties, and consequently can be difficult to learn with. Roderick et al. (2017) similarly constructs an abstract MDP like ours. However, due to critical design decisions, our approach outperforms theirs by nearly an order of magnitude. Whereas our approach monotonically grows the known set by saving worker parameters as transitions become reliable, Roderick et al. (2017) uses the same worker parameters to simultaneously learn many transitions. This causes catastrophic forgetting, as training on a new transition causes the worker to fail a previously learned transition, and prevents growth of the abstract MDP.

Only imitation learning methods (Aytar et al., 2018; Hester et al., 2018; Pohlen et al., 2018) outperform our method on the ALE's hardest exploration games. However, using demonstrations sidesteps the exploration problem our approach seeks to solve because following demonstrations leads to high reward.

## 8 DISCUSSION

This work presents a framework for tackling long-horizon, sparse-reward, high-dimensional tasks by using abstraction to decrease the dimensionality of the state space and to address compounding model errors. Empirically, this framework performs well in hard exploration tasks, and theoretically guarantees near-optimality. However, this work has limitations as well. First, our approach relies on some prior knowledge in the state abstraction function, although we compare against state-of-the-art methods using a similar amount of prior knowledge in our experiments. This information is readily available in the ALE, which exposes the RAM, and in many robotics tasks, which expose the underlying state (e.g., joint angles and object positions). Still, future work could attempt to automatically learn the state abstraction or extract the abstraction directly from the visible pixels. One potential method might be to start with a coarse represention, and iteratively refine the representation by splitting abstract states whenever reward is discovered. Another limitation of our work is that our simple theoretical guarantees require relatively strong assumptions. Fortunately, even when these assumptions are not satisfied, our approach can still perform well, as in our experiments.

**Reproducibility** Our code is available at `https://github.com/anonymous`

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

| Hyperparameter | Value |
|---|---|
| Success weight $\lambda_1$ | (**1** (stochastic), 10, **100** (deterministic)) |
| New transition exploration goal weight $\lambda_2$ | **5000** |
| Abstract state exploration goal weight $\lambda_3$ | **-2000** |
| Discovery exploration horizon $T_d$ | **50** |
| Discovery visit threshold $N_{visit}$ | **500** |
| Discovery repeat action range | 1 to 10, **1 to 20**, 1 to 30 |
| Worker horizon $H_{worker}$ | (10, 15, 20, **30**, 45) |
| Skill failure tolerance $\delta$ | (**0.05**, 0.1) |
| Skill holding heuristic $R_{hold}$ | **4** |
| Maximum transition distance $d_{max}$ | **15** |
| Dynamics sliding window size $N_{transition}$ | **100** |
| Adam learning rate | **0.001** |
| Max buffer size of each skill | **5000** |
| Skill DQN target sync frequency | **75** |
| Skill batch size | **32** |
| Skill minimum buffer size | **50** |
| Gradient norm clipping | **3.0** |
| Count-based weight $\beta$ | **0.63** |
| Margin weight $\lambda$ | **0.5** |

Table 2: Table of all hyperparameters and the values used in the experiments.

## A  EXPERIMENT DETAILS

Following Mnih et al. (2015), the pixel concrete states are downsampled and cropped to $84$ by $84$ and then are converted to grayscale. To capture velocity information, the worker receives as input the past four frames stacked together. Every action is repeated $4$ times.

In addition, MONTEZUMA'S REVENGE and PITFALL! are deterministic by default. As a result, the manager deterministically navigates to the fringes of the known set by calling on the worker's deterministic, saved skills. To minimize wallclock training time, we save the states at the fringes of the known set and enable the worker to teleport to those states, instead of repeatedly re-simulating the entire trajectory. When the worker teleports, we count all the frames it would have had to simulate as part of the training frames. Importantly, this only affects wallclock time, and does not benefit or change the agent in any way. Notably, this does not apply to PRIVATE EYE, where the initial state is stochastically chosen from two similar possible states.

### A.1  HYPERPARAMETERS

All of our hyperparameters are only tuned on MONTEZUMA'S REVENGE. Our skills are trained with the Adam optimizer (Kingma & Ba, 2014) with the default hyperparameters. Table 2 describes all hyperparameters and the values used during experiments (bolded), as well as other values that we tuned over (non-bolded). Most of our hyperparameters were selected once and never tuned.

### A.2  STATE ABSTRACTION FUNCTION

In MONTEZUMA'S REVENGE, each abstract state is a (bucketed agent x-coordinate, bucketed agent y-coordinate, agent room number, agent inventory, current room objects, agent inventory history) tuple. These are given by the RAM state at indices 42 (bucketed by 20), 43 (bucketed by 20), 3, 65, and 66 respectively. The agent inventory history is a counter of the number of times the current room objects change (the room objects change when the agent picks up an object).

In PITFALL!, each abstract state is a (bucketed agent x-coordinate, bucketed agent y-coordinate, agent room number, items that the agent has picked up) tuple. These are given by the RAM state at indices 97 (bucketed by 20), 105 (bucketed by 20), 1, and 113 respectively.

In PRIVATE EYE, each abstract state is a (bucketed agent x-coordinate, bucketed agent y-coordinate, agent room number, agent inventory, agent inventory history, tasks completed by the agent) tuple. These are given by the RAM state at indices 63 (bucketed by 40), 86 (bucketed by 20), 92, 60, 72, and 93 respectively.

## A.3    SKILL TRAINING AND ARCHITECTURE

**Architecture.**    Our skills are represented as Dueling DDQNs (van Hasselt et al., 2016; Wang et al., 2016), which produce the state-action value $Q_{(s,s')}(x,a) = A_{(s,s')}(x,a) + V_{(s,s')}(x)$, where $A_{(s,s')}(x,a)$ is the advantage and $V_{(s,s')}(x)$ is the state-value function. The skills recover a policy $\pi_{K(s,s')}(a|x,(s,s'))$ by greedily selecting the action with the highest Q-value at each concrete state $x$.

The skill uses the standard architecture (Mnih et al., 2015) to represent $A_{(s,s')}(x,a)$ and $V_{(s,s')}(x)$ with a small modification to also condition on the transition $(s,s')$. First, after applying the standard ALE pre-processing, the skill computes the pixel embedding $e_x$ of the pixel state $x$ by applying three square convolutional layers with (filters, size, stride) equal to $(32,8,4)$, $(64,4,2)$, and $(64,4,2)$ respectively with rectifier non-linearities (Nair & Hinton, 2010), and applying a final rectified linear layer with output size 512. Next, the skill computes the transition embedding $e_{(s,s')}$ by concatenating $[e_r; e_{diff}]$ and applying a final rectified linear layer with output size 64, where:

- $e_r$ is computed as the cumulative reward received by the skill during the skill episode, represented as one-hot, and passed through a single rectified linear layer of output size 32.

- $e_{diff}$ is computed as $s' - s$ passed through a single rectified linear layer of output size 96.

Finally, $e_x$ and $e_{(s,s')}$ are concatenated and passed through a final linear layer to obtain $A_{(s,s')}(x,a)$ and $V_{(s,s')}(x)$.

To prevent the skill from changing rapidly as it begins to converge on the optimal policy, we keep a sliding window estimate of its success rate $p_{success}$. At each timestep, with probability $1 - p_{success}$, we sample a batch of $(x,a,r,x')$ tuples for transition $(s,s')$ from the replay buffer and update the policy according the DDQN loss function: $\mathcal{L} = ||Q_{(s,s')}(x,a) - \text{target}||_2^2$, where target $= (r + Q^{\text{target}}(x', \arg\max_{a' \in \mathcal{A}} Q_{(s,s')}(x',a')))$. Additionally, since the rewards are intrinsically given, the optimal Q-value is known to be between 0 and $R_{hold}$. We increase stability by clipping target between these values.

**Pixel blindness.**    In addition, some skills are easy to learn (e.g. move a few steps to the left) and don't require pixel inputs to learn at all. To prevent the skills from unnecessarily using millions of parameters for these easy skills, the worker first attempts to learn *pixel-blind* skills for simple transitions $(s,s')$ with $d(s,s') = 1$ (i.e. $(s,s')$ was directly observed by the manager). The pixel-blind skills only compute $e_{(s,s')}$ and pass this through a final layer to compute the advantage and value functions (they do not compute or concatenate with $e_x$). If the worker fails to learn a pixel-blind skill, (e.g. if the skill actually requires pixel inputs, such as jumping over a monster) it will later try to learn a pixel-aware skill instead.

**Epsilon schedule.**    The skills use epsilon-greedy exploration, where at each timestep, with probability $\epsilon$, a random action is selected instead of the one produced by the skill's policy (Watkins, 1989). Once a skill becomes frozen, $\epsilon$ is permanently set to 0.

The number of episodes required to learn each skill is not known in advance, since some skills require many episodes to learn (e.g. traversing a difficult obstacle), while other skills learn in few episodes (e.g. moving a little to the left). Because of this, using an epsilon schedule that decays over a fixed number of episodes, which is typical for many RL algorithms, is insufficient. If epsilon is decayed over too many episodes, the simple skills waste valuable training time making exploratory actions, even though they've already learned near-optimal behavior. In contrast, if epsilon is decayed over too few episodes, the most difficult skills may never observe reward, and may consequently fail to learn. To address this, we draw motivation from the doubling trick in online learning Auer et al. (1995) to create an epsilon schedule, which accomodates skills requiring varying number of episodes to learn. Instead of choosing a fixed horizon, we decay epsilon over horizons of exponentially

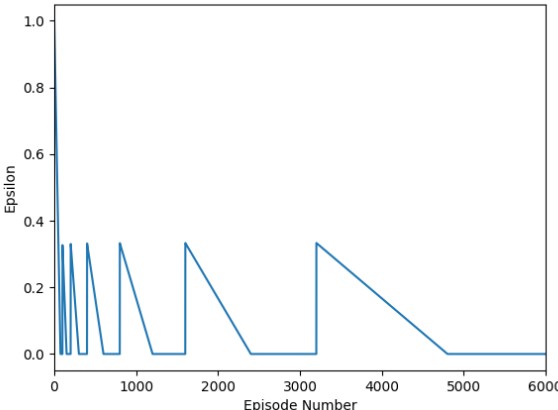

Figure 5: The saw-tooth epsilon schedule used by our skills

increasing length, summarized in Figure 5. This enables skills that learn quickly to achieve low values of epsilon early on in training, while skills that learn slowly will later explore with high values of epsilon over many episodes.

**Count-based exploration.**   Our skill additionally use count-based exploration similar to Tang et al. (2017); Bellemare et al. (2016) to learn more quickly. Each skill maintains a count of the number of times visit($s$) it has visited each abstract state $s$. Then, the skill provides itself with additional intrinsic reward to motivate itself to visit novel states, equal to $\frac{\beta}{\sqrt{\text{visit}(s)}}$ each time it visits abstract state $s$. We choose $\beta = 0.63$, an intrinsic reward of approximately $\frac{2}{\sqrt{10 \times \text{visit}(s)}}$.

**Self-imitation.**   When learning to traverse difficult obstacles (e.g. jumping over a disappearing floor), the skill may observe a few successes long before successfully learning a policy to reliably traverse the difficult obstacle. We use a variant of the self-imitation described in (Oh et al., 2018) to decrease this time. Whenever a skill successfully traverses a transition, it adds the entire successful trajectory to a separate replay buffer and performs imitation learning on the successful trajectories. These successful trajectories are actually optimal skill trajectories because the skill episode uses undiscounted reward, so all successful trajectories are equally optimal. To update on these skills, the skill periodically samples from this replay buffer and updates on an imitation loss function $\mathcal{L}_{imitation}(\theta) = \mathcal{L}_1(\theta) + \mathcal{L}_2(\theta)$, where $\theta$ is the skill's parameters, and $\mathcal{L}_1$ and $\mathcal{L}_2$ are defined as below:

- Let $G_t = \sum_{i=t}^{T} r_t$ be the reward to-go for a successful trajectory $(x_0, a_0, r_0), \cdots, (x_T, a_T, r_T)$. $\mathcal{L}_1$ directly regresses $Q_{(s,s')}(x, a)$ on the reward to-go of the successful trajectory, because $G_t$ is actually the optimal Q-value on successful trajectories (all successful trajectories are equally optimal): i.e., $\mathcal{L}_1 = ||G_t - Q_{(s,s')}(x_t, a_t)||_2$.

- We use the margin-loss from Hester et al. (2018) for $\mathcal{L}_2$. When sampling a transition $(x, a_E, r, x')$, $\mathcal{L}_2 = \max_{a \in \mathcal{A}}[Q_{(s,s')}(x, a) + \lambda \mathbf{1}[a = a_E]] - Q_{(s,s')}(s, a_E)$. Intuitively, $\mathcal{L}_2$ encourages the skill to replay the actions that led to successful trajectories over other actions. We use $\lambda = 0.5$, which was chosen with no hyperparameter tuning.

## B   ADDITIONAL RESULTS

### B.1   SKILL SHARING

The worker learns skills that successfully apply in to many similar transitions. Figure 6 depicts the number of different transitions each skill is used on in MONTEZUMA'S REVENGE, PITFALL!, and

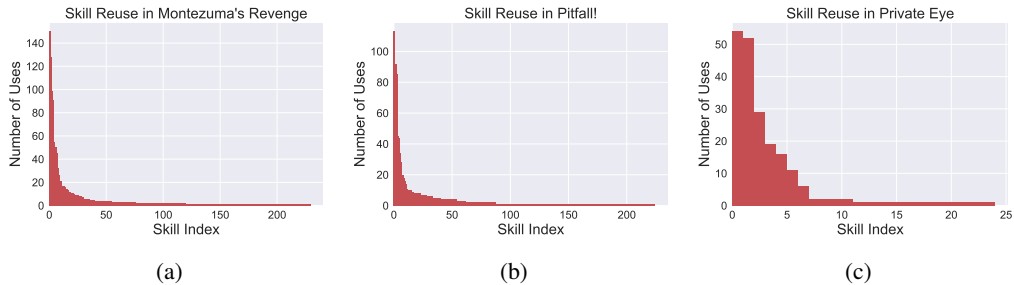

(a)             (b)             (c)

Figure 6: The number of different transitions each skill can traverse. Skills are sorted by decreasing usage.

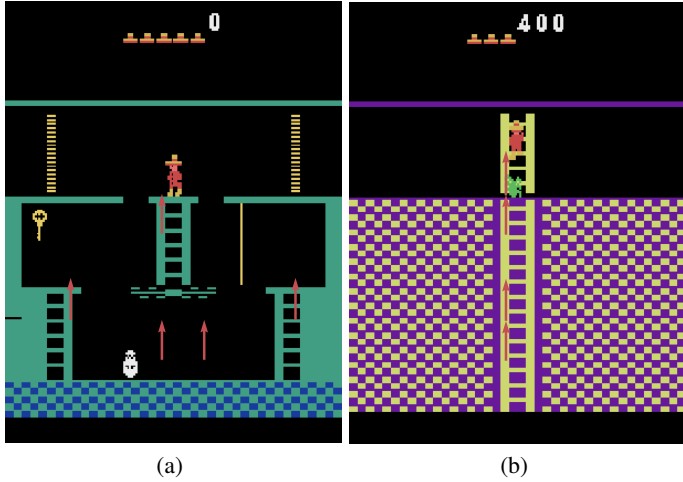

(a)                  (b)

Figure 7: Skill reuse in MONTEZUMA'S REVENGE. The same skill is useful in multiple rooms and can both climb up ladders and jump over a monster.

PRIVATE EYE. The simplest skills (e.g. move to the left) enjoy the highest number of reuses, while more esoteric skills (e.g. jump over a particular monster) are only useful in few scenarios.

Figure 7 provides an example of a skill in MONTEZUMA'S REVENGE with relatively high reuse. The arrows denote the movement of the agent when it executes the skill. The same skill that jumps over a monster in the first room (Figure 7(a)) can also climb up ladders. In Figure 7(b), the skill appears to know how to climb up all parts of the ladder except for this middle. This occurs because the spider occasionally blocks the middle of the ladder, and a different special skill must be used to avoid the spider. However, the skill reuse is not perfect. For example, in Figure 7(a), the skill can climb up the top half of ladders, but a separate skill climbs the bottom half of the ladders.

### B.2 GENERALIZATION TO NEW TASKS

To evaluate the ability of our approach to generalize to new reward functions, we train our approach on the basic MONTEZUMA'S REVENGE reward function and then test it on three challenging new reward functions (illustrated in Figure 8), not seen during training:

- *Get key:* the agent receives 1000 reward for picking up the key in room 14 (6 rooms away from the start). In addition, the agent receives -100 reward for picking up any other objects or opening any other doors.

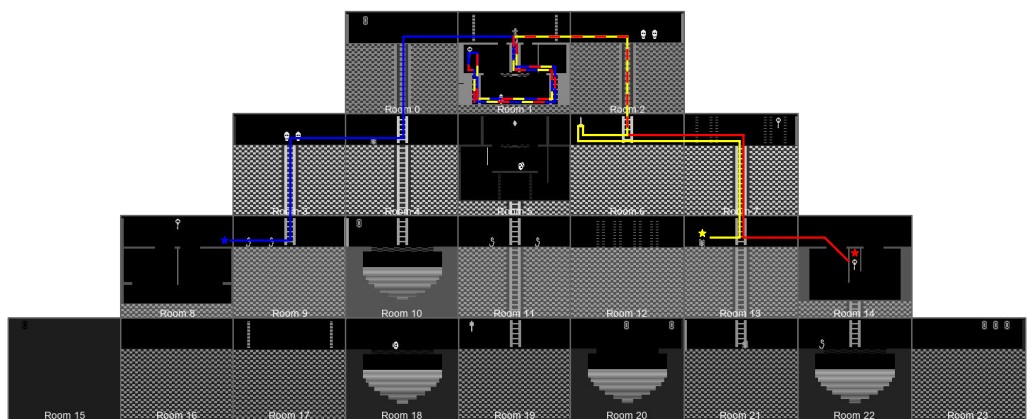

Figure 8: Display of all the rooms of the MONTEZUMA'S REVENGE pyramid. The agent starts in room 1 and must navigate through the pyramid, picking up objects and dodging monsters to complete the new tasks. The end of each task is marked with a star. Example paths for each task are marked with different colors. Multiple colors indicate sections of the paths that are shared across multiple tasks. (Red: *Get key*, Yellow: *Kill spider*, Blue: *Enter room 8*).

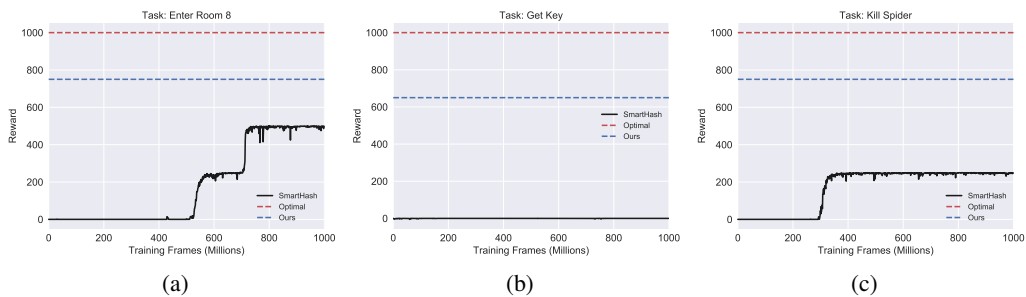

Figure 9: Training curves of *SmarthHash* on alternate tasks in MONTEZUMA'S REVENGE, compared with the performance of our approach generalizing to the new task. Our approach only receives 1M frames with the new task.

- *Kill spider:* the agent receives 1000 reward for killing the spider in room 13 (5 rooms away from the start). To kill the spider, the agent must first pick up the sword in room 6 (3 rooms away from the start) and save the sword for the spider. The agent receives no other reward.

- *Enter room 8:* the agent receives 1000 reward for entering room 8 (6 rooms away from the start). The agent receives no other reward.

In all three tasks, the episode ends when the agent completes its goal and receives positive reward.

Our approach trains on the basic MONTEZUMA'S REVENGE reward function for 2B frames, and then is allowed to observe the new reward functions for only 1M frames. We compare with *SmartHash*, which trains directly on the new reward functions for 1B frames. The results are summarized in Figure 9. Even when evaluated on a reward function different from the reward function it was trained on, our approach achieves about 3x as much reward as *SmartHash*, which is trained directly on the new reward function. Averaged over all 3 tasks, our approach achieves an average reward of 716.7 out of an optimal reward of 1000, whereas *SmartHash* only achieves an average reward of 220, even when trained directly on the new reward function. These experiments suggest that after our approach is trained in an environment on one task, it can quickly and successfully adapt to new tasks with little additional training.

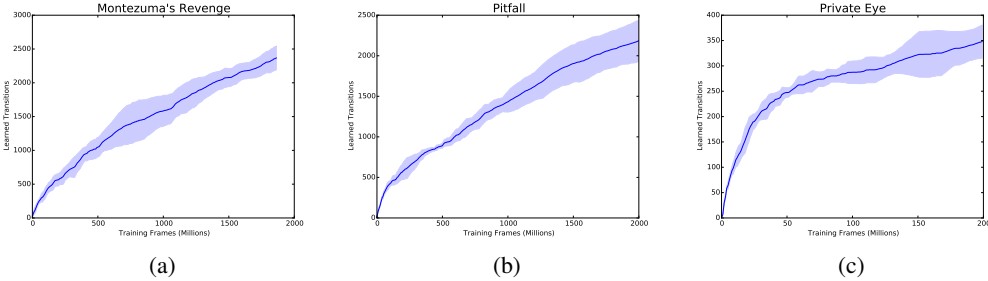

(a)          (b)          (c)

Figure 10: The number of transitions learned by the worker vs. number of training frames. The worker continues to learn new transitions even late into training, showing almost no signs of slowing down in MONTEZUMA'S REVENGE and PITFALL!.

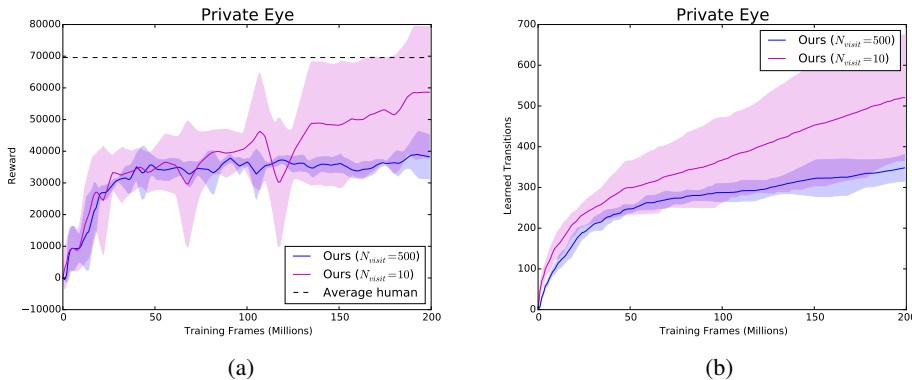

(a)          (b)

Figure 11: Performance of our approach on PRIVATE EYE, when decreasing the number of exploration episodes ($N_{visit}$). (a) When $N_{visit}$ is set to 10, our approach performs even better, achieving near human-level rewards. (b) Decreasing $N_{visit}$ enables the worker to learn more transitions in fewer frames.

### B.3    NEAR-LINEAR TRAINING

Whereas many prior state-of-the-art approaches tend to plateau toward the end of training, our approach continues to make near-linear progress. Figure 10 graphs the number of transitions learned by the worker against the number of frames in training. In MONTEZUMA'S REVENGE and PITFALL! particularly, the rate the worker learns new transitions is nearly constant throughout training. Because of this, when we continued to train a single seed on PITFALL!, by 5B frames, it achieved a reward of 26000, and by 20B frames, it achieved a reward of 35000.

### B.4    ADDITIONAL RESULTS ON PRIVATE EYE

By changing a single hyperparameter, our approach can perform even better on PRIVATE EYE, exceeding human performance on 2 of 4 seeds. Since nearby abstract states are particularly easy to discover in PRIVATE EYE, the manager needs not explore for new transitions as many times. Consequently, if we decrease the number of times the manager explores around each abstract state $N_{visit}$ from the 500 (used in the main experiments for all three games) to 10, performance improves. Figure 11 compares the performance with the decreased value of $N_{visit}$ with the original value of $N_{visit}$ reported in the main experiments. Decreasing $N_{visit}$ prevents the manager from wasting frames with unnecessary exploration and consequently enables the worker to learn more transitions in fewer total frames. With $N_{visit}$ set to 10, our approach achieves a final average performance of 60247 after 200M frames of training. Additionally, the top 2 of our 4 seeds achieve rewards of 75600 and 75400, exceeding average human performance: 69571 (Pohlen et al., 2018).

## C  GUARANTEES FOR NEAR OPTIMALITY

In general, a hierarchical policy over skills is not guaranteed to be near-optimal, because certain optimal trajectories may be impossible to follow using the skills. Because of this, hierarchical reinforcement learning literature typically focuses on hierarchical optimality (Dietterich, 2000) optimality given the abstractions. However, under the following assumptions, our approach provably achieves a near-optimal policy on the original MDP in time polynomial in the size of the abstract MDP, with high probability.

**Notation.**  Recall that we are interested in finding a near-optimal policy on the concrete MDP, with concrete states $x \in \mathcal{X}$ and concrete actions $a \in \mathcal{A}$. We refer to the value function of the optimal policy in the concrete MDP as $V^*(x)$.

From the concrete MDP approach constructs the abstract MDP, consisting of abstract states $s$ in the known set $\mathcal{S}$, learned abstract actions $o$ (e.g., go$(s, s')$), transition dynamics $P(s'|o, s)$, and rewards $R(s, s')$. The abstract MDP changes over time. We refer to the known set at timestep $t$ as $\mathcal{S}_t$ and to the set of all abstract states as $\Phi = \{\phi(x) : x \in \mathcal{X}\}$. We refer to the optimal value function on the abstract MDP at timestep $t$ as $V_t^*(s)$.

Our approach maintains estimates of the rewards and transition dynamics of the abstract MDP. With these estimates, at each timestep $t$, our approach computes $\pi_t$, the policy that is optimal with respect to these models (e.g., via value iteration). To simplify notation, we refer to the expected reward achieved by $\pi_t$ on the abstract MDP at timestep $t$ starting at abstract state $s$ as $V_t^{\pi_t}(s)$. The policy computed by our approach, $\pi_t$ also applies on the concrete MDP, because actions on the abstract MDP are implemented as subpolicies on the concrete MDP. Consequently, we refer to the expected reward achieved by $\pi_t$ on the concrete MDP starting at concrete state $x$ as $V^{\pi_t}(x)$.

**Assumptions.**  Formally, we require the following assumptions:

1. The learned abstract MDP is deterministic.

2. The learned abstract MDP has rewards that are *path independent*: i.e., all trajectories to an abstract state $s$ achieve the same reward.

3. The diameter of each abstract state is at most $H_{worker}$, where we define the diameter of an abstract state $s$ to be the maximum number of steps required to navigate to an immediate neighbor $s'$.

Assumption 1 intuitively says that the worker can successfully abstract away stochasticity from the manager, which our experiments in Section 6.3 suggests is possible. Humans typically also make this assumption when they plan. For example, when humans plan (e.g., to get to Paris), they expect to deterministically hit subgoals (e.g., get to the airport, get on the plane, get to the hotel) even though the world is actually non-deterministic (e.g., the taxi may be late.)

Assumption 2 tends to hold under many natural abstraction functions. For example, in the ALE games we evaluate on, the state abstraction function captures the agent's inventory and a history of the agent's inventory. Since all reward in these games is given when the agent picks up new items, or uses an item in its inventory, the agent's inventory and history encodes path independent reward. This also holds for many robotic arm manipulation tasks. For example, in a block stacking task with sparse rewards, a natural state abstraction might be the location of all the blocks. Then, the reward of a trajectory is encoded by the last abstract state of the trajectory: all trajectories that lead to a stacked configuration of blocks achieve the same reward for a success, while all non-stacking trajectories achieve the same failure reward.

Assumption 3 ensures that the worker has enough timesteps to navigate to any immediate neighbors. This is easily satisfied by setting $H_{worker}$ conservatively.

**Main results.**  While the above assumptions enable us to prove near-optimality, our method performs well empirically even when these assumptions are violated. Given the above assumptions, our main theoretical result holds:

**Proposition 1.** *Under the assumptions, for a given input $\eta$ and $\epsilon$, $\pi_t$ is at most $\epsilon$ suboptimal, $V^{\pi_t}(x_0) \geq V^*(x_0) - \epsilon$, on all but the first $O\big(|\Phi|^3(|\mathcal{A}| + \frac{\log K|\Phi| + \log \frac{1}{\eta}}{\log \frac{1}{p}}) + d_{max} \times H_{worker}\big)$ timesteps, where $x_0$ is the starting concrete state and $p$ and $K$ are polynomial in $|\Phi|$ and $|\mathcal{A}|$.*

To prove Proposition 1, we require the following three lemmas:

**Lemma 1.** *By setting $N_{transition}$ to be $O(\frac{\log 1 - (1-\eta')^{\frac{1}{|\Phi|^2}}/2}{2(\epsilon/|\Phi|HV^*(x_0))^2})$, with probability $1 - \eta'$, at each timestep $t$, $\pi_t$ is near-optimal on the current abstract MDP: i.e., $V_t^{\pi_t}(s) \geq V_t^*(s) - \epsilon$ for all abstract states $s \in \Phi$.*

**Lemma 2.** *If the known set is equal to the set of all abstract states $(\mathcal{S} = \Phi)$ at timestep $T$, then for any policy $\pi$ on the abstract MDP, $\pi$ achieves the same expected reward on the abstract MDP as on the concrete MDP: i.e., $V_T^\pi(\phi(x_0)) = V^\pi(x_0)$, where $x_0$ is the initial concrete state.*

*In addition, the expected return of the optimal policy on the abstract MDP is equal to the expected return of the optimal policy on the concrete MDP: i.e., $V_T^*(\phi(x_0)) = V^*(x_0)$ where $x_0$ is the initial state.*

**Lemma 3.** *With probability $1 - \eta$, the known set grows to cover all abstract states in $O\big(|\Phi|^3(|\mathcal{A}| + \frac{\log K|\Phi| + \log \frac{1}{\eta}}{\log \frac{1}{p}}) + d_{max} \times H_{worker}\big)$ time.*

Given these lemmas, we are ready to prove Proposition 1:

*Proof of Proposition 1.* For simplicity, we ignore terms due to appropriately setting $N_{transition}$ and $1 - \eta'$ from Lemma 1, but these terms are all polynomial in the size of the abstract MDP.

By Lemma 3 the known set grows to cover all abstract states in $T = O\big(|\Phi|^3(|\mathcal{A}| + \frac{\log K|\Phi| + \log \frac{1}{\eta}}{\log \frac{1}{p}}) + d_{max} \times H_{worker}\big)$ timesteps. For all timesteps $t \geq T$, by Lemma 1, $\pi_t$ is at most $\epsilon$ suboptimal on the abstract MDP. On all those timesteps, the known set is equal to all abstract states, so by Lemma 2, $\pi_t$ is at most $\epsilon$ suboptimal on the concrete MDP. □

**Proofs.** Now, we prove Lemma 1, Lemma 2, and Lemma 3.

*Proof of Lemma 1.* Let $\hat{P}(s'|o, s)$ denote the estimated transition dynamics[3] and $\hat{R}(s, s')$ denote the estimated reward model in the abstract MDP.

For each reliable transition $(s, s')$ (action in the abstract MDP), the manager estimates $\hat{P}(s'|o, s)$ from $N_{transition}$ samples of the worker. We bound the error in the model $|\hat{P}(s'|o, s) - P(s'|o, s)|$ with high probability by Hoeffding's inequality:

$$P(|\hat{P}(s'|o, s) - P(s'|o, s)| \geq \alpha) \leq 2e^{-2N_{transition}\alpha^2} \tag{1}$$

By the Assumption 2, $\hat{R}(s, s') = R(s, s')$ because all trajectories leading to $s$ achieve some reward $r$ and all trajectories leading to $s'$ achieve some reward $r'$, so a single estimate $R(s, s') = r' - r$ is sufficient to accurately determine $\hat{R}$.

Because the model errors are bounded, and because the abstract MDP is Markov, we can apply the simulation lemma (Kearns & Singh, 2002), which states that if $|\hat{P}(s'|o, s) - P(s'|o, s)| \leq \alpha$ and $|\hat{R}(s, s') - R(s, s')| \leq \alpha$, then the policy $\pi$ optimizing the MDP formed by $\hat{P}$ and $\hat{R}$ is at most $\epsilon$ suboptimal: i.e., at each timestep $t$, $V_t^\pi(s) \geq V_t^*(s) - \epsilon$ for all $s \in \mathcal{S}$, where $\alpha$ is $O\big((\epsilon/|\mathcal{S}|HV^*(x_0))^2\big)$, and $H$ is the horizon of the abstract MDP. Since the total number of transitions is bounded by $|\mathcal{S}|^2$, substituting for $N_{transition}$ gives the desired result. □

---

[3] In Section 2, we simplify notation to estimate the success rate instead of the full dynamics, but the manager could have estimated the full dynamics as required here.

*Proof of Lemma 2.* Assume that the known set is equal to the set of all abstract states at timestep $T$ and let $\pi$ be a policy on the abstract MDP.

To prove the first part of Lemma 2, consider a trajectory $s_0, o_0, r_0, s_1, o_1, \cdots, o_{T-1}, r_{T-1}, s_T$ rolled out by $\pi$ on the abstract MDP. Each abstract action $o_i$ is implemented as a subpolicy in the concrete MDP, so it expands to the trajectory solving the subtask of navigating from $s_i$ to $s_{i+1}$: $x_{(i,0)}, a_{(i,0)}, r_{(i,0)}, x_{(i,1)}, \cdots, o_{(i,T_i-1)}, r_{(i,T_i-1)}, x_{(i+1,0)}$, where $\phi(x_{(i,0)}) = s_i$, $\phi(x_{(i+1,0)}) = s_{i+1}$, and $r_i = \sum_{j=0}^{T_i-1} r_{(i,j)}$, by definition of the abstract MDP rewards. Consequently, for each trajectory, $\pi$ achieves the same total reward in the concrete MDP as in the abstract MDP, implying $V_T^\pi(\phi(x_0)) = V^\pi(x_0)$.

To prove the second part of Lemma 2, it suffices to show that the optimal policy on the concrete MDP achieves no more reward than $V_T^*(\phi(x_0))$, because the first part already shows that $V_T^*(\phi(x_0)) \leq V^*(x_0)$. Let $\pi^*$ be the optimal policy on the concrete MDP and let $\tau = x_0, x_1, \cdots, x_T$ be the highest-reward trajectory generated $\pi^*$ achieving reward $R$. Because the known set contains all abstract states, in particular, it contains $\phi(x_T)$. By Assumption 1, and because $\phi(x_T)$ is in the known set, it is possible to deterministically navigate to $\phi(x_T)$. By Assumption 2, traversing to $\phi(x_T)$ in the abstract MDP achieves reward $R$. Hence, $V_T^*(\phi(x_0))$ is at least $R$, proving the desired result. $\qquad\square$

*Proof of Lemma 3.* For the known set to cover all abstract states, the manager must discover all neighboring transitions for each abstract state and the worker must learn all the discovered transitions. The number of samples to do this is equal to the sum of:

1. The number of samples used by the manager to discover all transitions.

2. The number of samples used by the worker to learn new transitions.

3. The number of samples used by the worker to navigate to the fringes of the known set, for the worker to learn new transitions and for the manager to discover new transitions.

**Samples used by the manager to discover transitions.** At each abstract state, let $p$ be the probability that the manager fails to discover a particular abstract state on a single discovery episode. Let $K$ be the maximum number of outgoing transitions from an abstract state (maximum degree). Both $p$ and $K$ are polynomial in $|\Phi|$ and $|\mathcal{A}|$ because the diameter of each abstract state is bounded by assumption. By setting the number of times the manager explores from each abstract state, $N_{visit} = \frac{\log K|\Phi| + \log \frac{1}{\eta}}{\log \frac{1}{p}}$, the manager finds all outgoing transitions of each abstract state with probability at least $1 - \eta$ by the following elementary argument. There are at most $K|\Phi|$ total transitions to discover, and the manager fails to discover each transition with probability $p^{N_{visit}}$. By the union bound, the probability the manager fails to discover at least one transition is at most $K|\Phi|p^{N_{visit}}$. Consequently, the manager explores for $O(\frac{\log K|\Phi| + \log \frac{1}{\eta}}{\log \frac{1}{p}})$ timesteps.

**Samples used by the worker to learn transitions.** We assume that policy search with neural network function approximators can learn each transition at least as quickly as brute-force search over deterministic policies. By Assumption 3, the maximum number of timesteps required to traverse a transition $(s, s')$ is $d(s, s')$ times the diameter $H_{worker}$, which is at most $H = d_{max} \times H_{worker}$. Consequently, we bound the time required to learn each transition by $|\mathcal{A}|^H$, the total number of possible action trajectories for the worker.

**Samples used to navigate to the fringes of the known set.** The total number of samples used to navigate to the fringes of the known set is given by:

$$O(\sum_{s \in \Phi} N(s)) \tag{2}$$

where $N(s)$ is the number of times the worker visits state $s$ at the endpoint of a transition, counted when all abstract states are in the known set.

We now prove:

$$N(s) \leq N_{visit} + |E(G_s)|(|\mathcal{A}|^H + N_{visit}), \tag{3}$$

where $E(G_s)$ is the set of transitions in the subgraph of the abstract MDP consisting of directed transitions from $s$. This holds by strong induction on $|E(G_s)|$. In the base case, when $|E(G_s)| = 0$, $s$ is only visited $N_{visit}$ times for the manager to explore. In the inductive case, suppose $|E(G_s)| = c$ and that the inductive hypothesis holds for all values less than $c$. $N(s)$ is at most $N_{visit} + \sum_{(s,s') \in E(G_s)} N(s')$. Since $E(G_s) = \bigcup_{(s,s') \in E(G_s)} E(G_{s'}) \cup \{(s,s')\}$, the inductive hypothesis holds for each $N(s')$.

We bound $|E(G_s)|$ by the total number of transitions: $|\Phi|^2$. Substituting into (3) and (2), yields that total time for the worker to traverse to the fringes of the known set is: $O\big(|\Phi|^3(|\mathcal{A}| + \frac{\log K |\Phi| + \log \frac{1}{\eta}}{\log \frac{1}{p}})\big)$.

**Total samples.** Adding all three terms together gives that with probability at least $1 - \eta$, the known set covers all abstract states in $O\big(|\Phi|^3(|\mathcal{A}| + \frac{\log K |\Phi| + \log \frac{1}{\eta}}{\log \frac{1}{p}}) + d_{max} \times H_{worker}\big)$ samples. $\square$

## D    DISCOVERING NEW TRANSITIONS PSEUDOCODE

---
**Algorithm 3** DISCOVERTRANSITIONS($x_0$)

---
**Input:** called at concrete state $x_0$ to explore transitions near $\phi(x_0)$
 1: Choose $a_0 \sim \pi^d(x_0)$
 2: **while** $n(\phi(x_0)) \leq N_{visit}$ **do**
 3:    **for** $t = 1$ to $T_d$ **do**
 4:        Observe $x_t, r_t$
 5:        **if** $(\phi(x_{t-1}), \phi(x_t))$ has never been observed before **then**
 6:            Update reward model $R(\phi(x_{t-1}), \phi(x_t)) \leftarrow r_t$
 7:            Add transition to transitions model $P(\phi(x_{t-1}), \phi(x_t)) \leftarrow 0$.
 8:        Choose $a_t \sim \pi^d(x_{1:t}, a_{1:t-1})$
 9:    Continue exploring: reset $x_0 \leftarrow x_{T_d}$ and choose $a_0 \sim \pi^d(x_0)$

---

