# OpenReview forum: "Learning Abstract Models for Long-Horizon Exploration"
_ICLR.cc/2019/Conference_

### Official Review · AnonReviewer3 · 2018-11-02
**The proposed algorithm outperforms the state of the art algorithms on three very hard games**

**Rating:** 6
**Confidence:** 2

**Review:**

This paper considers reinforcement learning tasks that have high-dimensional space, long-horizon time, sparse-rewards. In this setting, current reinforcement learning algorithms struggle to train agents so that they can achieve high rewards. To address this problem, the authors propose an abstract MDP algorithm. The algorithm consists of three parts: manager, worker, and discoverer. The manager controls the exploration scheduling, the worker updates the policy, and the discoverer purely explores the abstract states. Since there are too many state, the abstract MDP utilize the RAM state as the corresponding abstract state for each situation.

The main strong point of this paper is the experiment section. The proposed algorithm outperforms all previous state of the art algorithms for Montezuma’s revenge, Pitfall!, and Private eye over a factor of 2.

It is a minor weak point that the algorithm can work only when the abstract state is obtained by the RAM state. In some RL tasks, it is not allowed to access the RAM state.

================================
I've read all other reviewers' comments and the response from authors, and decreased the score. Although this paper contains interesting idea and results, as other reviewers pointed out, it is very hard to compare with other algorithm. I agree to other reviewers. The algorithm assumptions are strong.

---

> ### Author Response · Authors · 2018-11-15
> **Reply to Reviewer 3**
>
> We thank Reviewer 3 for their comments. Reviewer 3 points out the strong state-of-the-art performance of our approach as a strength and mentions prior knowledge (our use of RAM state information) as a minor weakness. To clarify, in our experiments, we outperform previous non-demonstration state-of-the-art approaches that use a comparable amount of prior knowledge. We discuss our usage of prior knowledge in greater detail in the section titled “Prior Knowledge” in our response to Reviewer 2.

---

### Official Review · AnonReviewer2 · 2018-11-04
**Effective but complex method which achieves good exploration performance conditioned on substantial prior knowledge**

**Rating:** 5
**Confidence:** 4

**Review:**

This paper considers how to effectively perform exploration in the setting where a difficult, high-dimensional MDP can be mapped to a simpler, lower-dimensional MDP. They propose a hierarchical approach where a model of the abstract MDP is incrementally learned, and then used to train sub-policies to transition between abstract states. These sub-policies are trained using intrinsic rewards for transitioning to the correct state, and the transition probabilities in the abstract MDP reflect how well a sub-policy can perform the transition.

The approach is evaluated on three difficult Atari games, which all require difficult exploration: Montezuma's Revenge, Pitfall and Private Eye, and is shown to achieve good performance in all of them. Furthermore, the model can be used to generalize to new tasks by changing the rewards associated with different transitions.

The main downside with this paper is that the mapping from original state (i.e. pixels) to the abstract state is assumed to be known beforehand, which requires prior knowledge. The authors hardcode this mapping for each of the games by fetching the relevant bits of information from RAM. This prevents fair comparison to many other methods which only use pixels, and makes this paper borderline rather than strong accept.


Quality: the method is evaluated on difficult problems and shown to perform well. The experiments are thorough and explore a variety of dimensions such as robustness to stochasticity, granularity of the abstract state and generalization to new tasks. The approach does strike me as rather complicated though - it requires 19 (!) different hyperparameters as shown in table 2. The authors do mention that many of these did not require much tuning and they intend on making their code public. Still, this suggests that re-implementation or extensions by others may be challenging. Are all of these moving parts necessary?

Clarity: the paper is well-written, for the most part clear, and the details are thoroughly described in the appendix.

Originality: this approach in the context of modern deep learning is to my knowledge novel.

Significance: This paper provides a general approach for hierarchical model-based planning when the mapping from the hard MDP to the easy one is known, and in this sense is significant. It is limited by the assumption that the mapping to abstract states is known. I suspect the complexity of the approach may also be a limiting factor.

Pros:
+ good results on 3 challenging problems
+ effective demonstration of hierachical model-based planning

Cons:
- requires significant prior knowledge for state encoding
- complicated method

Minor:
- in the intro, last paragraph: "Our approach significantly outperforms previous non-demonstration SOTA approaches in all 3 domains". Please specify that you use extra knowledge extracted from RAM, otherwise this is misleading.
- Algorithm 1: nagivate -> navigate
- Section 4, last sentence: broken appendix link.
- Bottom of page 6: "Recent work on contextual MDPs...as we do here" is not a sentence.
- In related work, it would be nice to mention some relevant early work by Schmidhuber on subgoal generation: http://people.idsia.ch/~juergen/subgoals.html



*** Updated ***

After reading the updated paper, responses, other reviews, and looking at related works more closely, I have changed my score to a 5. This is due to several factors.

Although the paper's core idea is definitely interesting, the fact that they use hardcoded features, rather the standard setup which uses pixels, makes comparison to other methods much more complicated. In particular, I think that the comparison to DQN-PixelCNN is unfair, as this other method makes very few assumptions about the inputs (only that they are pixels). The authors sort of point this out in the main text, but this is somewhat misleading. They say "PixelCNN uses less prior knowledge than our approach". In fact, it uses as much prior knowledge as any RL method which operates on pixels. Granted, this is nonzero, but it's vastly less than what this paper's method assumes. The other comparison is to SOORL (which uses a different state encoding altogether). The comparison to SmartHash is fairer, although the variant of SmartHash they compare against is not the main method the paper proposes (a generic autoencoder-based state encoding which makes minimal assumptions about the input). It would have been better if the authors included experiments for their method using such a learned state encoding.

Reporting SOTA results on very hard tasks using extra hardcoded features or other domain knowledge is potentially misleading to the community as to how far along we are in solving these tasks, and extra care should be taken to put these results in context. Otherwise, for those not familiar with the subtleties, this makes it seem like these tasks are being solved when in fact they are not. My concern is that other works may then be asked to be compared against these artificially high results. Having many different task setups also makes comparison between different published works confusing in general. Other works (such as Ostrovski et al) have been able to make progress on these tasks while staying within the standard pixel-based framework.

These concerns would have been partially mitigated had the authors made it *very* clear that they were assuming substantial prior knowledge, which makes their method non-comparable to others which do not make this assumption. This could have been done in the introduction (which was one of my comments, but this was not included in the updated draft). I.e., something to the effect of "We emphasize that our approach assumes substantially more prior knowledge than other approaches which operate only on pixels, and as such is not directly comparable with these approaches". In addition, I would have liked if the authors had followed the suggestion of Reviewer 1 to include results in pixel space, even if negative, but this was not done either (using a simple autoencoder-based representation, like the one in the SmartHash paper, would have also been fine). As it is, statements such as "Our approach achieves more than 2x the reward of prior non-demonstration SOTA approaches" and "our approach relies on some prior knowledge in the state abstraction function, although we compare against SOTA methods using a similar amount of prior knowledge in our experiments" are quite misleading and unfair to other methods which do not assume access to prior knowledge (the second statement is untrue for the case of DQN-PixelCNN).

Another point which I had not noticed previously is the very high sample complexity (2 billion). One of the motivations behind model-based approaches is that they are supposed to be more sample efficient, but that does not seem to be the case here.

---

> ### Author Response · Authors · 2018-11-15
> **Reply to Reviewer 2 [1 / 2]**
>
> We would like to thank Reviewer 2 for their detailed and thoughtful feedback! Reviewer 2 raises two main concerns: 1) that our approach requires prior knowledge and 2) that our approach is complicated, which we address in the two sections below:
>
> ----------------------------------------
>
> Prior Knowledge
>
> In this work, we assume access to prior knowledge (i.e., RAM state information) in the form of the state abstraction function. However, in our experiments, we compare with state-of-the-art approaches that use a comparable amount of prior knowledge (these approaches use more prior knowledge in 1 game, the same prior knowledge in 1 game, and less prior knowledge in 1 game). In each game, we compare with the highest scoring non-demonstration approach and we achieve new state-of-the-art results in each game, by over 2x:
>
> - In Montezuma’s Revenge, we compare with SmartHash, which requires RAM state information equivalent to the prior
> knowledge used by our approach. Our approach achieves over 2x as much reward as SmartHash on average.
> - In Pitfall!, we compare with SOORL, which requires parsing out all the relevant objects on the screen, prior knowledge much stronger than that used by our approach. Our approach achieves over 10x as much reward on average. In addition, we also compare with Apex DQfD, which uses expert demonstrations, even stronger prior knowledge. Our approach achieves about 2.5x the reward of Apex DQfD on average. We note that no prior approach has ever achieved >0 reward on Pitfall! with only RAM state information (our approach achieves ~10K reward).
> - In Private Eye, we compare with DQN-CTS, which encodes the prior knowledge that semantically different states tend to have very different pixels. DQN-CTS uses weaker prior knowledge than our approach, but we compare with DQN-CTS because it achieves the best performance out of all non-demonstration prior approaches. Our approach achieves over 2x as much reward as DQN-CTS on average.
>
> To further understand what portion of the performance of our method is due to just prior knowledge, we’ve run additional experiments with AbstractStateHash, an approach (described in greater detail in the paper) which uses the same prior knowledge as our approach and uses this prior knowledge to do count-based exploration (count-based exploration methods have achieved the prior state-of-the-art results in the hardest exploration games). In the initial submission, we already reported results of AbstractStateHash on Montezuma’s Revenge, which achieves results competitive with the prior state-of-the-art; our approach achieves >2x the reward of AbstractStateHash. We will soon submit an updated draft with results of AbstractStateHash on Pitfall! and Private Eye and we provide a summary below.
>
> - On Pitfall!, AbstractStateHash achieves 0 reward (comparable with many strong approaches, e.g., DQN-PixelCNN and Rainbow), whereas our approach achieves ~10K reward.
> - On Private Eye, our approach achieves >100x the reward of AbstractStateHash.
>
> These results suggest that while the RAM state prior knowledge does provide our approach valuable signal, prior state-of-the-art methods do not effectively leverage this prior knowledge.
>
> In addition, in Section 7.5, we analyze the effect of varying the state abstraction function to answer the question of: how hard is it to find a state abstraction function that works well with our method? We find that our approach significantly outperforms the prior state-of-the-art under many abstract state representations. This alleviates the burden of selecting the perfect state abstraction function for new tasks (in our case, for each game, we selected an abstraction function and never changed or tuned it), and suggests that future work could find different state abstraction functions requiring less prior knowledge. In other domains, it may also be possible to easily extract abstract states from the state. For example, many robotics tasks have fully observable states (e.g. consisting of joint angles of a robotic arm and positions of objects). In these tasks, a good state abstraction function might just extract the dimensions corresponding to the position of the gripper and the positions of the objects.

---

> > ### Author Response · Authors · 2018-11-15
> > **Reply to Reviewer 2 [2 / 2]**
> >
> > Complexity
> >
> > While our approach has many pieces, it consists of three highly modularized components with simple interfaces: the manager, worker, and discoverer. These components can be (and in our case were) developed and improved separately, significantly limiting the effective complexity of working with the system. For example, the worker can use any state-of-the-art RL algorithm to learn its goal-conditioned policy. In addition, in contrast to most end-to-end deep RL methods whose metrics (e.g., Q-values, loss functions) are hard to interpret, the metrics in the framework are interpretable and make debugging and improving the system easier. For example, the growth of the safe set indicates good exploration, and the number of episodes required for the worker to learn each transition indicates how well the worker’s RL algorithms are learning. We plan to release our code to further aid reproducibility efforts.
> >
> > Reviewer 2 notes that our approach has many (19) hyperparameters. We used the same hyperparameters to achieve state-of-the-art performance on all games and only tuned (exclusively on Montezuma’s Revenge) 4 hyperparameters total, suggesting that applying our approach to new tasks may not require heavy hyperparameter tuning. In addition, while our approach does have many hyperparameters, the total number of hyperparameters is comparable to other approaches, e.g. DQN-CTS has 14 hyperparameters.
> >
> > ----------------------------------
> >
> > Minor
> >
> > We thank Reviewer 2 for pointing out these minor issues and will address them in newer drafts, which we will post shortly.

---

### Official Review · AnonReviewer1 · 2018-11-12
**Relevant topic, poor evaluation, unclear related work**

**Rating:** 4
**Confidence:** 4

**Review:**

This paper deal with learning abstract MDPs for planning in tasks that require long-horizon due to sparse rewards.
This is an extremely important and timely topic in the RL community.

The paper is generally clear and well written.

The proposed algorithm seems reasonable and it is conceptually simple to understand. In the current experimental results presented it also seems to outperform the alternative baselines.

Nonetheless, the paper has few flaws that significantly impact the stated contributions and reduced my rating.
1) a stated contribution are theoretical guarantees about the performance of the algorithm. this analysis is not currently included in the main body of the manuscript, but rather in the appendix, which I find rather annoying. Moreover, said the analysis is in my opinion not sufficiently rigorous, with hand-wavy arguments, no formal proof and unclear terms (e.g. how do you define near-optimal?). Moreover, as observed by the authors this analysis currently rely on strong assumptions that might make it rather unrealistic. Overall, if you want to claim theoretical guarantees you will have to significantly improve the manuscript.
2) Related work, although extensive in terms of the number of references, do not help to place this work in the literature. Listing related work is no the same as describing similarities and differences compared to previous methods. For example, a paper that obviously comes to mind is "FeUdal Networks for Hierarchical Reinforcement Learning". What are the differences to your approach? Also, please place the related work earlier on in the paper. Otherwise, it is impossible for a reader to correctly and objectively relate your proposed approach to previous literature.
3) In its current form, the experimental results are extremely cherry-picked, with a very small number of tasks evaluated, and for each task a single selected baseline used. This needs to be changed: a) you should run all the baselines for each of the current tasks b) you should also expand the experiments evaluated to include tasks where it is not obvious that a hierarchy would help/is necessary c) you should include more baselines. feudal RL should be one, Roderick et al 2017 should be another one (especially considering your discussion in Sec 8)

Additional feedback:
- The paper is currently oriented towards discrete states. What can you say about continuous spaces?
- The use of random exploration for the discoverer is underwhelming. Have you tried different approaches? Would more advanced exploration techniques work or improve the performance?
- Using only 4 seeds seems too little to provide accurate standard deviations. Please run at least 10 experiments.
- The use of RAM is a fairly serious limitation of your experimental setting in my view. You should include results also for the pixel space, even if negative. Otherwise, this choice is incomprehensible.

---

> ### Author Response · Authors · 2018-11-27
> **Reply to Reviewer 1 [1/2]**
>
> We thank Reviewer 1 for their detailed comments and feedback. Reviewer 1’s main concerns are 1) that the related works section does not sufficiently frame our work with previous literature, 2) that the proofs of theoretical guarantees are not sufficiently rigorous, and 3) that the experiments section is not comprehensive enough. We have posted a significantly updated new draft to address these concerns.
>
> -------------------------------------
>
> Experiments
>
> Reviewer 1 claims that we do not sufficiently compare with enough other methods, and specifically asks for comparisons with Feudal Networks (FuN) and Roderick et al., 2017. We already comprehensively compare with the prior non-demonstration state-of-the-art, which use a comparable amount of prior knowledge, in each game. Since we already compare with the prior state-of-the-art approaches, and other approaches perform significantly worse than the prior state-of-the-art approaches, we do not compare with the many other deep RL approaches. In particular, FuN and Roderick et al., 2017 both report results on Montezuma’s Revenge. The prior state-of-the-art approach we compare against, SmartHash, outperforms these approaches by 1.75x and 4x respectively, at the number of frames they report (200M and 50M respectively). Our approach further outperforms SmartHash by over 2x.
>
> Reviewer 1 further asks for evaluation on more games. We believe that we have already demonstrated a significant improvement over the prior state-of-the-art, and additional experiments could be prohibitively expensive. In particular, we follow Aytar et al., 2018, and evaluate on 3 of the hardest exploration games from the Arcade Learning Environment. We do not evaluate on many of the simpler other games (e.g., Breakout), because they do not require sophisticated exploration and can already be solved with current state-of-the-art methods. We use the same set of minimally tuned hyperparameters (tuned only on Montezuma’s Revenge) and obtain new state-of-the-art results by over 2x, suggesting that our approach can generalize to new tasks. Our results are not cherry-picked as R1 suggests: following many recent deep RL works, e.g., Ostrovski et al., 2017, Tang et al., 2017, we run 4 seeds on each task, and obtain statistically significant results. Even our *worst seed* outperforms or is competitive with the prior state-of-the-art *best seed*.
>
> We note that running 10 seeds would approximately cost $30,000 per additional game in compute. Renting the appropriate equipment (e.g., via Google Cloud) to run a single seed to completion costs ~$1,500. To run 20 seeds (10 for our approach, 10 for the prior state-of-the-art) would cost 20 x $1,500 = $30,000 or roughly the median US annual salary.
>
> ---------------------------------------
>
> Related Works
>
> We’ve updated the related works section in our recently posted draft to more carefully compare  Please see Sections 1 and 7 for updated related work. The main critical difference between our work and other HRL works is that we build an abstract MDP, which enables us to plan for targeted exploration; other works also learn skills and operate in latent abstract state spaces, but not necessarily in a way that satisfies the property of an MDP, which can make effectively using the learned skills difficult.
>
> --------------------------------------
>
> Theory
>
> In the updated draft of our paper, we have updated the rigor of the theory section: please see Section 5 and Appendix C for updated theory. To summarize: we’re interested in the sample complexity of RL algorithms, i.e., the number of samples required for the learned policy to become near-optimal (achieve reward at most epsilon less than the optimal policy). Standard results (e.g., MBIE-EB, R-MAX) can guarantee a near-optimal policy, but they require so many samples (polynomial in the size of the state space) in deep RL settings, that the guarantees are effectively vacuous. In contrast, for a subclass of MDPs, our approach provably learns a near-optimal policy in a number of samples polynomial in the size of the *abstract* MDP.

---

> > ### Author Response · Authors · 2018-11-27
> > **Reply to Reviewer 1 [2/2]**
> >
> > Responding to R1's additional feedback:
> >
> > R1 asks if our method applies to continuous spaces. Our method applies to continuous spaces with no changes, we can just discretize the abstract state (not the concrete state). In particular, our method may be well-suited for many robotics tasks, which often have the full state (e.g., joint angles and object positions) available. For example, in a task like stacking blocks with a robotic arm, a good state abstraction function would be the position of the end effector and blocks, which are directly available in the state (e.g., in Stacker from DM Control Suite).
> >
> > R1 says that the randomized exploration used by the discoverer is underwhelming. We view the simplicity of the discoverer as advantageous. Fundamentally, exploration requires some degree of randomness, and we were already able to achieve state-of-the-art results without overcomplicating the discoverer. We note that this random exploration is only for locally discovering nearby abstract states. Globally, we drive exploration by incrementally growing the safe set (renamed known set in the updated draft).
> >
> > R1 asks for experiments that do not use RAM state information. We clarify that we use the RAM state information for the state abstraction function, which is a fundamental component of our work, so it is not possible to run experiments without this RAM information. However, we explore the robustness of our method to the exact chosen abstraction in section 7.4 and find that our method achieves state-of-the-art results over a wide range of state abstraction functions, suggesting that alternate state abstraction functions could be used. We also note that our experiments compare with state-of-the-art approaches, which also use prior knowledge comparable to our usage of RAM state information.

---

> > > ### Public Comment · (anonymous) · 2018-11-27
> > > **Why not compare to Ostrovski et al. in Montezuma's Revenge?**
> > >
> > > Benchmarking against SmartHash on Montezuma's Revenge seems a bit arbitrary to me, given that Ostrovski et al. also reported a maximum score of 6,600 for their best seed and they *don't* exploit access to the game's RAM. (The assumptions that they do make are certainly not "comparable" to assuming RAM access).
> > >
> > > I'm not saying this is definitely the case, but choosing to cite SmartHash instead looks like a deliberate attempt to support your argument about prior knowledge, especially given that you're aware of Ostrovski et al.'s work. At the very least, you should give Ostrovski et al. credit for reaching 6,600 too.

---

> > > > ### Author Response · Authors · 2018-11-27
> > > > **Response**
> > > >
> > > > Thanks for the comment. We compare with the non-demonstration approach achieving the highest *mean* reward, averaged across multiple seeds, because an algorithm's *max* reward across multiple seeds is typically not fully representative of its performance (Henderson et al., 2017). While DQN-PixelCNN (Ostrovski et al., 2017) and DQN-CTS (Bellemare et al., 2016) match the *max* performance of SmartHash, which we will note in future drafts, they perform much worse on *average*. At the 150M frames reported by these works, SmartHash achieves a mean reward of 4645, while DQN-CTS achieves a mean reward of 3705, and DQN-PixelCNN achieves a mean reward of 2514.

---

> > > > > ### Public Comment · (anonymous) · 2018-11-28
> > > > > **Probably worth mentioning training time too**
> > > > >
> > > > > Thanks for the response, making that change will alleviate most of my concern. As a side note, I just noticed that in Figure 2 your Montezuma agents have been trained for 2 billion frames of experience. For fairness, you should probably note that this is about 10x more experience than Ostrovski et al. and Bellemare et al.'s agents were trained on. (Or expand the table to include results after 150m frames if you're going to put the figures side-by-side.) There's a worrying trend in recent work on hard exploration games to amp up the training time without mentioning it. (The RND paper that appeared concurrently with this one underplays sample efficiency too.)

---

> > > > > > ### Author Response · Authors · 2018-11-29
> > > > > > **Our approach is relatively sample efficient**
> > > > > >
> > > > > > We are definitely sympathetic to this concern and will note this in future drafts. Importantly, our approach is *more sample efficient* than the prior state-of-the-art (SOTA), and achieves SOTA results when compared with other approaches at 150M frames (the number of frames used by Ostrovski, et al., 2017).
> > > > > >
> > > > > > Concretely, our approach achieves higher reward than the prior SOTA at every point along the training curves in Montezuma’s Revenge and Private Eye, except from ~85M to ~95M frames of training, where SmartHash achieves roughly the same reward as our approach on Montezuma's Revenge. Directly comparing sample complexity with SOORL (Keramati et al., 2018) on Pitfall is not possible, because they save samples by manually specifying an extremely simplified model class and manually extracting all objects.
> > > > > >
> > > > > > Specifically at 150M frames of training:
> > > > > >     - On Montezuma's Revenge, our approach achieves a mean reward of 4875,
> > > > > >        compared to SmartHash (4645), DQN-CTS (3705), and DQN-PixelCNN (2514).
> > > > > >     - On Pitfall, our approach achieves a mean reward of 332 compared to 80 from
> > > > > >        SOORL. Notably, no prior approaches have achieved positive reward on Pitfall
> > > > > >        without prior knowledge much stronger than ours (e.g., demonstrations or
> > > > > >        knowing a simplified model class and extracting all objects as in SOORL).
> > > > > >     - On Private Eye, our approach achieves a mean reward of 35897, compared to DQN-
> > > > > >       PixelCNN (15806). Note that DQN-PixelCNN achieves a mean reward of 15806 in the
> > > > > >       middle of training; by 150M frames, its performance actually drops to 7787.
> > > > > >
> > > > > > Reporting results on more frames enables us to differentiate our approach, which continues to learn even after 150M frames of training, from other approaches that plateau (e.g., SmartHash, DQN-PixelCNN):
> > > > > >     - On Montezuma's Revenge, SmartHash barely improves after 150M frames,
> > > > > >       achieving only 5001 reward after 2B frames of training. In contrast, our
> > > > > >       approach continues to learn and achieves a mean reward of 11020.
> > > > > >     - On Pitfall, our approach surpasses the average reward of a strong learning from
> > > > > >       demonstrations approach, ApeX DQfD (Pohlen et al., 2018), by ~800M frames
> > > > > >       (~4000 reward). By ~1.6B frames, our approach surpasses human performance
> > > > > >       (6464) and achieves a mean reward of 9959 after 2B frames of training. Out of
> > > > > >       curiosity, we ran a single seed on Pitfall for even longer. This single seed achieved a
> > > > > >       reward of 29000 by 5B frames of training, and a reward of 35000 by 20B frames of
> > > > > >       training. We did not run multiple seeds for so many frames due to computational
> > > > > >       resources.
> > > > > >     - On Private Eye, our approach improves more slowly. However, if we change a single
> > > > > >       hyperparameter (Appendix B.4), our approach achieves a mean reward of >60000
> > > > > >       by 200M frames of training, nearing superhuman performance. All other results are
> > > > > >       from running the same set of minimally tuned hyperparameters across all games,
> > > > > >       where those hyperparameters were exclusively tuned on Montezuma's Revenge.
> > > > > >
> > > > > > Finally, our theoretical results (Section 5 and Appendix C) provide sample complexity guarantees that require exponentially fewer samples than prior algorithms with sample complexity guarantees (e.g., R-MAX, MBIE-EB).

---

### Author Response · Authors · 2018-12-01
**Summary of Updated Draft**

We thank all the reviewers for their feedback. We uploaded a significantly improved draft at the end of the rebuttal period to incorporate the reviewer’s feedback. Here is a summary of the main changes:

1) To measure the significance of our prior knowledge (RAM state information), we evaluated AbstractStateHash on all 3 games (details in Section 6). AbstractStateHash (a variant of SmartHash) combines the prior state-of-the-art method of count-based exploration with the exact RAM state information used by our approach. While AbstractStateHash performs comparably to the prior state-of-the-art on two games, it performs poorly on one and is outperformed by our approach by >2x on each game. This suggests that the RAM state information does not trivialize the games and that prior state-of-the-art methods do not effectively leverage this information.

2) We simplified the presentation of our algorithm to highlight the key ideas (Section 2, 3, 4). The key idea behind our approach is to construct the abstract Markov Decision Process (MDP), a representation of the task (the concrete MDP) meeting several critical properties:
    - Its state space is low-dimensional, making it computationally tractable to plan (e.g., via value iteration). Planning
      enables our approach to perform targeted exploration.
    - We maintain accurate estimates of its transition dynamics and rewards, enabling us to plan without compounding
      errors.
    - At any point in time, it is an accurate representation on a subset of the abstract states, and eventually, our
      approach grows it to cover all abstract states. This enables our approach to learn a high-reward policy on the
      abstract MDP, which is simple because the abstract MDP is so small. Then we use that policy to obtain a
      high-reward policy on the concrete MDP.
We incrementally grow the abstract MDP by learning a worker policy, which learns the subtask of navigating between pairs of nearby abstract states in the abstract MDP.

3)  We updated our related works (Section 7) to better compare with the hierarchical reinforcement learning (HRL) literature and Roderick et al., 2017. The key difference between our work and many prior HRL works is our construction of the abstract MDP. While other HRL works also operate in latent abstract state spaces and learn skills / options, they do not enforce that the latent abstract state space forms a MDP with the learned skills, which prevents them from exploiting properties of MDPs as our approach does: i.e., planning and avoiding exponentially many state histories. Roderick et al., 2017 also forms an abstract MDP like our approach, but it differs on crucial design decisions, (e.g., how to grow the abstract MDP), which causes it to perform nearly an order of magnitude worse than our approach (see Section 7 for details).

4)  We improved the rigor of our theoretical results (Section 5 and Appendix C). Our results concern sample complexity: the number of samples required to learn a near-optimal policy with high-probability. Prior algorithms (e.g., R-MAX) provably guarantee learning a near-optimal policy, but require so many samples that the guarantee is vacuous. In contrast, for a subclass of MDPs, our approach provably learns a near-optimal policy in an exponentially smaller number of samples.

---

### Meta-Review · Area_Chair1 · 2018-12-18
**innovative approach and strong results, concerns about comparison to baselines**

**Confidence:** 4
**Recommendation:** Reject

**Metareview:**

The paper presents a novel approach to exploration in long-horizon / sparse reward RL settings. The approach is based on the notion of abstract states, a space that is lower-dimensional than the original state space, and in which transition dynamics can be learned and exploration is planned. A distributed algorithm is proposed for managing exploration in the abstract space (done by the manager), and learning to navigate between abstract states (workers). Empirical results show strong performance on hard exploration Atari games.

The paper addresses a key challenge in reinforcement learning - learning and planning in long horizon MDPs. It presents an original approach to this problem, and demonstrates that it can be leveraged to achieve strong empirical results.

At the same time, the reviewers and AC note several potential weaknesses, the focus here is on the subset that substantially affected the final acceptance decision. First, the paper deviates from the majority of current state of the art deep RL approaches by leveraging prior knowledge in the form of the RAM state. The cause for concern is not so much the use of the RAM information, but the comparison to other prior approaches using "comparable amounts of prior knowledge" - an argument that was considered misleading by the reviewers and AC. The reviewers make detailed suggestions on how to address these concerns in a future revision. Despite initially diverging assessments, the final consensus between the reviewers and AC was that the stated concerns would require a thorough revision of the paper and that it should not be accepted in its current stage.

On a separate note, a lot of the discussion between R1 and the authors centered on whether more comparisons / a larger number of seeds should be run. The authors argued that the requested comparisons would be too costly. A suggestion for a future revision of the paper would be to only run a large number (e.g., 10) of seeds for the first 150M steps of each experiment, and presenting these results separately from the long-running experiments. This should be a cost efficient way to shed light on a particularly important range, and would help validate claims about sample efficiency.